# Unsupervised 3D Object Learning through Neuron Activity aware Plasticity

**Beomseok Kang, Biswadeep Chakraborty & Saibal Mukhopadhyay**
School of Electrical and Computer Engineering
Georgia Institute of Technology, Atlanta, GA 30332, USA
`{beomseok, biswadeep, smukhopadhyay6}@gatech.edu`

## Abstract

We present an unsupervised deep learning model for 3D object classification. Conventional Hebbian learning, a well-known unsupervised model, suffers from loss of local features leading to reduced performance for tasks with complex geometric objects. We present a deep network with a novel **Ne**uron **A**ctivity A**w**are (NeAW) Hebbian learning rule that dynamically switches the neurons to be governed by Hebbian learning or anti-Hebbian learning, depending on its activity. We analytically show that NeAW Hebbian learning relieves the bias in neuron activity, allowing more neurons to attend to the representation of the 3D objects. Empirical results show that the NeAW Hebbian learning outperforms other variants of Hebbian learning and shows higher accuracy over fully supervised models when training data is limited.

## 1 Introduction

Supervised deep networks for recognizing objects from 3D point clouds have demonstrated high accuracy but generally suffer from poor performance when labeled training data is limited (Wu et al., 2015; Qi et al., 2017a;b; Wang et al., 2019; Maturana & Scherer, 2015). On the other hand, self-supervised or unsupervised models can be trained without labeled data hence improving the performance in data efficient scenarios. Self-supervised learning methods have been studied for 3D object recognition mostly in an autoencoder setting, which necessarily reconstructs input to learn the representation (Achlioptas et al., 2018; Girdhar et al., 2016). Unsupervised learning has also been applied to pre-process the input for an encoder but still largely relying on supervised learning (Li et al., 2018). Conventionally, self-organizing maps and growing neural gas have been used as fully unsupervised learning for 3D objects while they aim to reconstruct the surface of the objects (do Rêgo et al., 2007; Mole & Araújo, 2010). A fully unsupervised deep network for 3D object classification has rarely been studied.

Unsupervised Hebbian learning is known to offer attractive advantages such as data efficiency, noise robustness, and adaptability for various applications (Najarro & Risi, 2020; Kang et al., 2022; Miconi et al., 2018; Zhou et al., 2022). The basic Hebbian and anti-Hebbian learning refer to that synaptic weight is strengthened and weakened, respectively, when pre- and post-synaptic neurons are simultaneously activated (Hebb, 2005). Many past efforts have developed variants of Hebb's rule. Examples include Oja's rule and Grossberg's rule (Oja, 1982; Grossberg, 1976) for object recognition (Amato et al., 2019; Miconi, 2021), ABCD rule (Soltoggio et al., 2007) for meta-learning and reinforcement tasks (Najarro & Risi, 2020), and another variant for hetero-associative memory (Limbacher & Legenstein, 2020). However, Hebbian learning is often vulnerable to the loss of local features (Miconi, 2021; Bahroun et al., 2017; Bahroun & Soltoggio, 2017; Amato et al., 2019). This is a major challenge for applying Hebbian rules for tasks with more complex geometric objects, such as object recognition from 3D point clouds.

In this paper, we present an unsupervised deep learning model for 3D object recognition that uses a novel neuron activity-aware plasticity-based Hebbian learning to mitigate the vanishing of local features, thereby improving the performance of 3D object classification. We observe that, in networks trained with plain Hebbian learning, only a few neurons always activate irrespective of the object class. In other words, spatial features of 3D objects are represented by the activation of only a

few specific neurons, which degrades task performance. We develop a hybrid Hebbian learning rule, referred to as the **Neu**ron **A**ctivity **Aw**are (NeAW) Hebbian, that relieves the biased activity. The key concept of NeAW Hebbian is to dynamically convert the learning rule of synapses associated with an output neuron from Hebbian to anti-Hebbian or vice versa depending on the activity of the output neuron. The reduction of bias allows a different subset of neurons to activate for different object classes, which increases class-to-class dissimilarity in the latent space. Our deep learning model uses a feature extraction module trained by NeAW Hebbian learning, and a classifier module is trained by supervised learning. The feature extractor designed as a multi-layer perceptron (MLP) transforms the positional vector of sampled points on 3D objects into high-dimensional space (Qi et al., 2017a;b). The experimental results evaluated on ModelNet10 and ModelNet40 (Wu et al., 2015) show that the proposed NeAW Hebbian learning outperforms the prior Hebbian rules for efficient unsupervised 3D deep learning tasks. This paper makes the following key contributions:

- We present a deep learning model for 3D object recognition with the NeAW Hebbian learning rule that dynamically controls Hebbian and anti-Hebbian learning to relax the biased activity of neurons. The NeAW Hebbian learning efficiently transforms spatial features of various classes of 3D objects into a high-dimensional space defined by neuron activities.

- We analytically prove that the NeAW Hebbian learning relieves the biased activity of output neurons if the input is under a given geometric condition, while solely applying Hebbian or anti-Hebbian learning does not guarantee the relaxation of the skewed activity.

- We analytically prove that purely Hebbian learning and anti-Hebbian learning on the biased neuron activity leads to a poor subspace representation with few principal components, thereby limiting the performance in the classification tasks.

- We empirically demonstrate that the NeAW Hebbian learning rule outperforms the existing variants of Hebbian learning rules in the 3D object recognition task. We also show that NeAW Hebbian learning achieves higher accuracy than end-to-end supervised learning when training data is limited (data-efficient learning).

## 2 RELATED WORK AND BACKGROUND

**Deep Learning Models for 3D Object Recognition**   Supervised 3D convolutional neural network (CNN) models have been developed to process a volumetric representation of 3D objects (Maturana & Scherer, 2015; Wu et al., 2015) but high sparsity in volumetric input increases computation cost, limiting applications to low-resolution point clouds. Multi-view CNN renders 3D objects into images at different views and processes these images using 2D CNN (Su et al., 2015). VGG and ResNet-based models show good performance when the training images are well-engineered under proper multi-views (Su et al., 2018). However, 2D CNN-based approaches are difficult to scale to complex 3D tasks (Qi et al., 2017a). Recently low-complexity point-based models have been designed to process a point cloud where input is $(x, y, z)$ coordinates of points (Qi et al., 2017a). Self-supervised learning models have also been proposed based on autoencoder and generative adversarial networks (Wu et al., 2016; Sharma et al., 2016). Both models accept a voxel representation of 3D objects and learn the latent representation of the objects by reconstructing the voxel. The learned representation is used as the input to an additional classifier. Our unsupervised learning does not use labels, but unlike existing self-supervised models, our approach does not reconstruct the objects.

**Hebbian Learning Models**   The variants of the Hebbian learning rule are given as:

$$\mathbf{w}(t+1) = \begin{cases} \mathbf{w}(t) + \eta y \mathbf{x} & \text{Hebb's rule} \\ \mathbf{w}(t) + \eta y (\mathbf{x} - y \mathbf{w}(t)) & \text{Oja's rule} \\ \mathbf{w}(t) + \eta y (\mathbf{x} - \mathbf{w}(t)) & \text{Grossberg's rule} \end{cases} \tag{1}$$

where $\mathbf{x}$, $y$, and $\mathbf{w}$ are the input, output, and weight, $\eta$ is the learning rate. Hebb's rule is the basic form of Hebbian learning where weights are updated if both the input and output neuron fire (Hebb, 2005). This linear association can be interpreted as biologically plausible principal component analysis (PCA) if the data samples are assumed to be zero-mean (Weingessel & Hornik, 2000). However, the plain Hebb's rule is often vulnerable to the divergence of weight vectors as

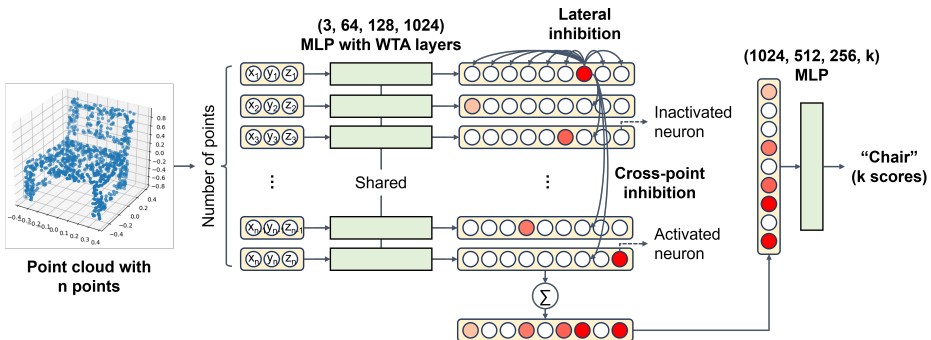

Figure 1: 3D object classification model.

there is no explicit upper bound (Porr & Wörgötter, 2007). Oja's rule has an additional constraint term on Hebb's rule to normalize the weight vectors (Oja, 1982). It is derived by dividing the weight update in plain Hebb's rule by its norm, assuming that the input and output are linearly associated. Grossberg's rule is another variant where a constraint term enables the weight to be converged on the input; hence the weight is more aligned with the frequently observed data (Grossberg, 1976). Recent works have used Grossberg's rule for image feature extraction (Miconi, 2021; Amato et al., 2019), We can also modify Grossberg's rule to be identical to Kohonen's self-organizing map (SOM) by applying Winner-Take-All (WTA) to output neurons and can be used for learning two-dimensional or three-dimensional spatial features (Kohonen, 2012; Li et al., 2018). Another variant of Grossberg's rule with the notion of neuron activity is recently studied for learning 2D point sets (Kang et al., 2022). In this paper, we mainly compare NeAW learning with Hebb's rule, Grossberg's rule, and Oja's rule and demonstrate the proposed rule is superior to them in the 3D object classification task.

## 3   PROPOSED APPROACH

### 3.1   3D OBJECT CLASSIFICATION MODEL

We process $(x, y, z)$ coordinates of points in a point cloud by a shared three-layer MLP with competition layers. NeAW learning is applied on the MLP (i.e. encoder), and the other layers (i.e. classifier) are trained by supervised learning. The output vector for a point is written by:

$$\mathbf{x}^{l_{i+1}} = \text{WTA}(\text{ReLU}(\mathbf{W}^{l_i,T}\mathbf{x}^{l_i})) \tag{2}$$

where $\mathbf{x}^{l_i}$, $\mathbf{W}^{l_i}$, and $\mathbf{x}^{l_{i+1}}$ are the input, weight, and output vector at layer $l_i$, respectively. Winner-take-all (WTA) is applied on the output of each layer so that the single output neuron with the highest similarity (i.e. the lowest $||\mathbf{x}^{l_i} - \mathbf{W}^{l_i}_{:,j}||$ for $\mathbf{x}^{l_{i+1}}_j$) has a non-zero value while others are forced to be zero (Coates et al., 2011; Hu et al., 2014; Miconi, 2021). Figure 1 shows the architecture of our model. There are two types of inhibition for the last output neurons in the encoder. Each WTA layer makes lateral inhibition, as shown in the figure, and cross-point inhibition is applied to the last output neurons after the lateral inhibition. Then, the model aggregates the transformed vector by summing the elements of the remaining neurons across the columns. The cross-inhibition and the summation can be interpreted as a MaxPooling layer that directly finds the maximum value of each column. The proposed model consequently arrives at a PointNet-style model which combines the MLP and MaxPooling layer in the encoder design. The aggregated activation of the last neurons, the global feature of objects, is processed by FC layers with ReLU and Softmax activation in the classifier. LayerNorm layers are included between the FC layers. We observe the WTA modules play an important role in the representation learning. The proposed learning in the model without the WTA modules is discussed in Appendix E.

### 3.2   MOTIVATIONAL STUDY

The main motivation of the paper is to demonstrate the unsupervised learning rule that can balance the neuron activity and its theoretical analysis in the 3D object classification task. Figure 2 shows

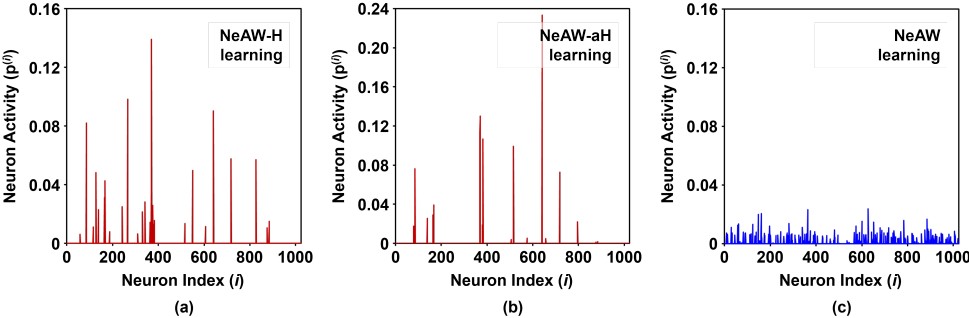

Figure 2: The average neuron activity of each output neuron for ModelNet10 with (a) NeAW learning with only positive weight update (Hebbian-like), (b) NeAW learning with only negative weight update (anti-Hebbian-like), and (c) NeAW learning.

the average neuron activity in the last layer of the encoder with different learning rules. Note, the details of the figure are explained again in Section 4. We measure how frequently each output neuron activates across different points, so-called neuron activity, to quantify whether the transformed vectors are distinguishable in high-dimensional space or merged to a simple global feature. We observe that having only positive or negative weight update in the NeAW Hebbian learning lead to few specific neurons in the last layer to frequently activate regardless of objects. Given input vectors are projected onto weight vectors, the biased activation of output neurons indicates the small number of principal weight vectors in high-dimensional space. In other words, multiple input points are associated with few weight vectors, which significantly decreases the variance of context vectors with different labels and consequently leads to classification failure. Hence, the skewed activity's relaxation is considered the key problem to solve in this paper.

### 3.3 Neuron Activity aware Hebbian Learning

**Definition 1.** *Neuron activity* $p^{(j)}$ of the $j$-th output neuron is the number of the activation divided by the number of entire points. The neuron activity is 1 if the neuron activates for all the points in a point cloud, and 0 if the neuron never activates.

**Definition 2.** *Activation boundary* is the plane with the same Euclidean distance from two weight vectors that determine which neuron of the weight vector to activate.

Our learning rule is given by:

$$\mathbf{W}_{:,j}(t+1) = \mathbf{W}_{:,j}(t) + f(p^* - p^{(j)})\frac{\eta}{N}\sum_{i=0}^{N-1}\mathbf{1}_{i \in \arg\min_{k}\|\mathbf{x}_k - \mathbf{W}_{:,j}(t)\|_2}(\mathbf{x}_i - \mathbf{W}_{:,j}(t)) \quad (3)$$

$$\mathbf{W}_{:,j}(t+1) = \mathbf{W}_{:,j}(t) + f(p^* - p^{(j)})\frac{\eta}{N}(\mathbf{x}_k - \mathbf{W}_{:,j}(t))$$
$$= \begin{cases} \mathbf{W}_{:,j}(t) + a\frac{\eta}{N}(\mathbf{x}_k - \mathbf{W}_{:,j}(t)) & \text{if } p^* > p^{(j)} \\ \mathbf{W}_{:,j}(t) - b\frac{\eta}{N}(\mathbf{x}_k - \mathbf{W}_{:,j}(t)) & \text{if } p^* < p^{(j)} \end{cases} \quad (4)$$

where $\mathbf{W}_{:,j}(t)$ is the weight vector of the $j$-th output neuron at $t$, $\mathbf{x}_i$ is the $i$-th input vector in a point cloud, $p^*$ is the optimal neuron activity. As WTA layers in the proposed model search the closest weight vectors from the input vectors, the learning rule is motivated by Grossberg's rule and the prior work (Kang et al., 2022). The optimal neuron activity is given by $1/d$ where $d$ is the number of output neurons (Kang et al., 2022). The indicator function is considered another WTA. Each weight vector searches the closest input vector with regard to Euclidean distance and is moved closer or further to the input vector depending on the range of $p^{(j)}$. It describes the conversion between Hebbian and anti-Hebbian learning using the indicator function $f$, indicating Hebbian learning for low-activity neurons ($p^* > p^{(j)}$) and anti-Hebbian learning for high-activity neurons ($p^* < p^{(j)}$). Note, the weight is not updated if the neuron activity is optimal. The learning rule is simplified to (4) if the $k$-th input vector is closest to the weight vector. We introduce a constant $a$ and $b$ in the equation to vary the importance of Hebbian and anti-Hebbian learning. In addition to the NeAW

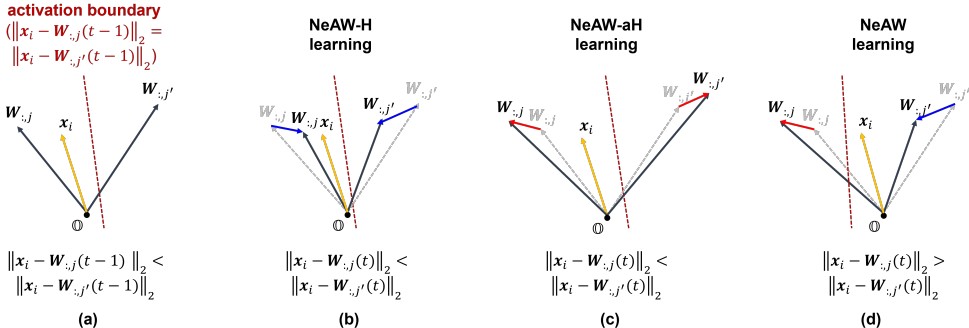

Figure 3: Geometry of input and weight vectors. (a) the initial geometry of the vectors and the changed geometry of the vectors after (b) NeAW-H learning, (c) NeAW-aH learning, and (d) NeAW learning.

Hebbian learning rule, we define NeAW-H learning (a=1 and b=-1) and NeAW-aH learning (a=-1 and b=1) by making the learning rule only has either positive or negative (i.e. non-hybrid) weight update regardless of the activity. Note, NeAW Hebbian learning does not refer NeAW-H learning. We observe that balancing the neuron activity is also an important function in the brain (Keck et al., 2017; Turrigiano & Nelson, 2004). The biological plausibility of the proposed learning rule is discussed in Appendix G.

An important question is how the hybridization of the learning rules depending on the neuron activity effectively relaxes the biased neuron activity. In a simplified scenario, we can consider two weight vectors and an input vector with the assumption that the corresponding neuron of each weight vector is highly active or non-active. Also, $a$ and $b$ are assumed to be 1. The problem is whether NeAW Hebbian learning shifts the activation boundary so that the other neuron is activated after learning.

To help the intuitive understanding of how NeAW Hebbian learning changes the activation boundary, the geometry of the three vectors is described in Figure 3. In Figure 3(a), the activation boundary as a dotted line shows that input $\mathbf{x}_i$ is left to the boundary plane, and the $j$-th neuron is assumed to be highly active while the $j'$-th neuron is not active before learning. We expect that the learning properly shifts the activation boundary to activate the $j'$-th neuron and relieve the biased activity. The geometry of the vectors after NeAW-H, NeAW-aH, and NeAW Hebbian learning are compared in Figure 3(b), (c), and (d), respectively. The activity-agnostic rules move both the weight vectors to be closer to or further from the activation boundary, implying the activation boundary cannot be effectively shifted to either left or right space. It is desirable that both the weight vectors are also shifted in the same direction. In this light, NeAW Hebbian learning offers the appropriate shift of the activation boundary and consequently switches the winner neuron from the highly active $j$-th neuron to the non-active $j'$-th neuron, as shown in Figure 3(d).

Based on this idea, we develop formal theorems and corollaries to mathematically prove that NeAW Hebbian learning changes the winner neuron while the others cannot:

**Theorem 1.** *Let two weight vectors $\mathbf{W}_{:,j}$ and $\mathbf{W}_{:,j'}$ with the biased activity and an input vector $\mathbf{x}_i$ satisfy $\|\mathbf{x}_i - \mathbf{W}_{:,j}(t)\|_2 < \|\mathbf{x}_i - \mathbf{W}_{:,j'}(t)\|_2$ at $t$. Then, NeAW Hebbian learning changes the sign of the inequality to $\|\mathbf{x}_i - \mathbf{W}_{:,j}(t+1)\|_2 > \|\mathbf{x}_i - \mathbf{W}_{:,j'}(t+1)\|_2$ at $t+1$ if the vectors are on the geometric condition $(1 + \frac{\eta}{N})\|\mathbf{x}_i - \mathbf{W}_{:,j}(t)\|_2 > (1 - \frac{\eta}{N})\|\mathbf{x}_i - \mathbf{W}_{:,j'}(t)\|_2$.*

*Proof Sketch.* (Formal proof in Appendix A) Due to the biased activity, each weight vector is updated by Hebbian learning or anti-Hebbian learning. Then, we have two update rules, $\mathbf{W}_{:,j}(t+1) = \mathbf{W}_{:,j}(t) - \frac{\eta}{N}(\mathbf{x}_i - \mathbf{W}_{:,j}(t))$ for the high-active $j$-th neuron and $\mathbf{W}_{:,j'}(t+1) = \mathbf{W}_{:,j'}(t) + \frac{\eta}{N}(\mathbf{x}_i - \mathbf{W}_{:,j'}(t))$ for the low-active $j'$-th neuron. From the assumption, we have $\|\mathbf{x}_i - \mathbf{W}_{:,j}(t)\|_2 < \|\mathbf{x}_i - \mathbf{W}_{:,j'}(t)\|_2$, which indicates that the winner neuron is the $j$-th neuron at $t$. Using the two updates rule to substitute the weight vectors from $t$ to $t+1$, the inequality at $t+1$ is derived. The sign in the inequality is flipped if the Euclidean distances satisfy $(1 + \frac{\eta}{N})\|\mathbf{x}_i - \mathbf{W}_{:,j}(t)\|_2 > (1 - \frac{\eta}{N})\|\mathbf{x}_i - \mathbf{W}_{:,j'}(t)\|_2$.

**Corollary 1.1.** *Let two weight vectors $\mathbf{W}_{:,j}$ and $\mathbf{W}_{:,j'}$ with the biased activity and an input vector $\mathbf{x}_i$ satisfy $\|\mathbf{x}_i - \mathbf{W}_{:,j}(t)\|_2 < \|\mathbf{x}_i - \mathbf{W}_{:,j'}(t)\|_2$ at $t$. Then, Hebbian learning (NeAW-H) preserves the sign of the inequality to $\|\mathbf{x}_i - \mathbf{W}_{:,j}(t+1)\|_2 < \|\mathbf{x}_i - \mathbf{W}_{:,j'}(t+1)\|_2$ at $t+1$.*

**Corollary 1.2.** *Let two weight vectors $\mathbf{W}_{:,j}$ and $\mathbf{W}_{:,j'}$ with the biased activity and an input vector $\mathbf{x}_i$ satisfy $\|\mathbf{x}_i - \mathbf{W}_{:,j}(t)\|_2 < \|\mathbf{x}_i - \mathbf{W}_{:,j'}(t)\|_2$ at $t$. Then, anti-Hebbian learning (NeAW-aH) preserves the sign of the inequality to $\|\mathbf{x}_i - \mathbf{W}_{:,j}(t+1)\|_2 < \|\mathbf{x}_i - \mathbf{W}_{:,j'}(t+1)\|_2$ at $t+1$.*

Proof for corollaries is in Appendix A.

The prior study has shown that Grossberg's rule combined with WTA does k-means clustering (Hu et al., 2014). While NeAW Hebbian learning does not aim to minimize the distance between the input and weight required for the clustering, we can interpret the activated neurons as cluster centroids. The more output neurons the model activates with the balanced activity, the smaller spatial features are responsible for each neuron if they create separable clusters. In other words, more activated neurons are preferred to avoid the vanishing of local features, which can be achieved by maximizing the variance of the neuron activity. The variance at layer $l_i$ is written by:

$$
\begin{aligned}
\mathrm{Var}(\mathbf{y}^{l_i}) &= \frac{1}{N} \sum_{k=0}^{N-1} (\mathbf{y}_k^{l_i} - \mu_{\mathbf{y}^{l_i}}) \cdot (\mathbf{y}^{l_i} - \mu_{\mathbf{y}^{l_i}}) \\
&= \frac{1}{N} \sum_{k=0}^{N-1} \mathbf{y}_k^{l_i} \cdot \mathbf{y}_k^{l_i} - \frac{1}{N^2} \sum_{k,w=0}^{N-1} \mathbf{y}_k^{l_i} \cdot \mathbf{y}_w^{l_i} = 1 - \frac{1}{N^2} \sum_{k,w=0}^{N-1} \mathbf{y}_k^{l_i} \cdot \mathbf{y}_w^{l_i}
\end{aligned}
\tag{5}
$$

where $y^{l_i}$ is a binary vector whose element is either 0 or 1. It is obtained from (2) by applying a step function on it only to consider whether the neuron is activated or not. Also, $y_k^{l_i} \cdot y_k^{l_i}$ is always 1 as the binary vector has a single non-zero element due to WTA. The proposed NeAW Hebbian learning eventually aims to increase the variance of the activated neurons so that each neuron is responsible for the different geometric parts of 3D objects by properly rotating the weight vectors through the neuron activity aware plasticity, or more formally:

**Theorem 2.** *For a network of $N$ neurons in the biased activity with weight vectors $\mathbf{W}$ and an input vector $\mathbf{x}$, the variance of the distribution of activated neurons follows $\mathrm{Var}(\mathrm{NeAW\ Hebbian}) \geq \mathrm{Var}(\mathrm{Hebbian})\,\mathrm{or}\,\mathrm{Var}(\mathrm{anti-Hebbian})$ which means NeAW Hebbian learning provides the greatest variance of the neuron activity compared to solely using Hebbian or anti-Hebbian learning.*

*Proof Sketch.* (Formal proof in Appendix A) First, we note that Hebbian (H) and anti-Hebbian (aH) networks optimize the information capacity by orthonormalizing the principal subspace (Plumbley, 1993). To prove that the neuronal activations for the hybrid learning rule have a greater variance than the standard H and aH learning rules, we first assume that a network with a greater number of neuronal activations can encode a large number of principal components in the subspace. We consider that the local H and aH learning rules recover a principal subspace from the data using a principled cost function (Pehlevan et al., 2015). Hence, we rewrite the problem as:

$$
\mathrm{span}_{\min}(\mathcal{L}_{NeAW}) \geq \mathrm{span}_{\min}(\mathcal{L}_H)\,\mathrm{or}\,\mathrm{span}_{\min}(\mathcal{L}_{aH})
\tag{6}
$$

where $\mathcal{L}_H$ and $\mathcal{L}_{aH}$ denote, the subspace learned using the Hebbian and anti-Hebbian learning rules, respectively. Let us consider the case where we have $N$ neurons learned using H or aH learning rules. Without loss of generality, we can assume that the first $n$ ($n < N$) neurons are trained using the learning method $\mathcal{L}$ and the metric space of the neuron vectors and the weight vectors to be represented as $\mathcal{W}$. For the $n+1$-th neuron, the neuron can be trained using either the same learning rule $\mathcal{L}$ or the complementary learning rule $\bar{\mathcal{L}}$. Thus, we study the differential gain from using one learning rule over the other for the $n+1$-th neuron. A learning rule extracts the principal components of the input space by projecting the input space into the orthogonal subspace, and a neural network with a larger number of principal components can learn a better representation of the input space. Again, H and aH learning rules lead to different principal components (Rubner & Tavan, 1989; Carlson, 1990). Hence, we show that using NeAW Hebbian learning approach leads to an increase in the number of the principal components learned, which leads to a greater span of the learned subspace for the hybrid learning model compared to the homogeneous H or aH learning method, which, in turn, entails a higher variance for the neuronal activity.

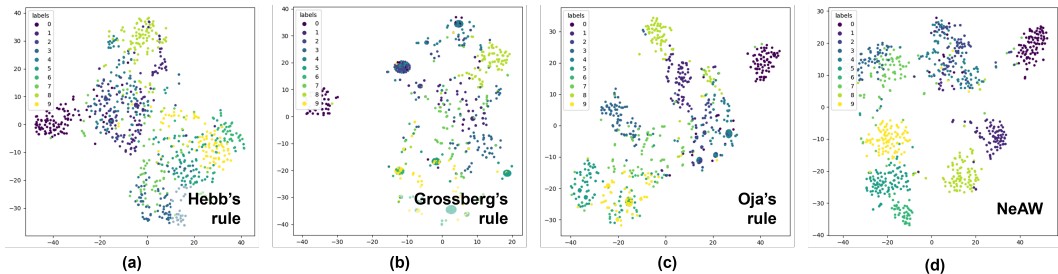

Figure 4: t-SNE for ModelNet10 in (a) Hebb's rule, (b) Grossberg's rule, (c) Oja' rule, and (d) NeAW learning.

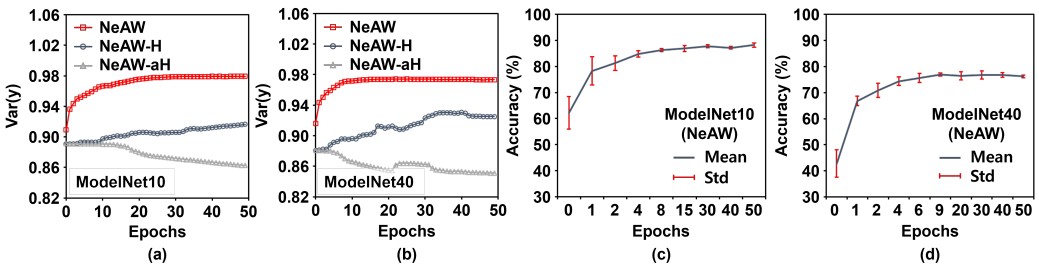

Figure 5: Effect of training with regard to the variance of neuron activity and test accuracy. (a) the variance of neuron activity with the different learning rules in ModelNet10 and (b) ModelNet40 and (c) the test accuracy at different epochs for ModelNet10 and (d) ModelNet40.

## 4 EXPERIMENTAL RESULT

**Experimental setting** The proposed model is evaluated on ModelNet10 and ModelNet40 datasets, which include 10-class and 40-class 3D CAD objects for 3D deep learning (Wu et al., 2015). We randomly sample 1024 points on the surface of the objects. The encoder is trained by unsupervised learning with unlabeled datasets while the classifier is trained by supervised learning with object labels. Note, the weights of the encoder are not changed during training the classifier. For all the unsupervised learning rules, we set training epoch 50, learning rate 1e-2, and train batch 4; however, learning rate is proportionally increased when the amount of training data is limited. For example, learning rate is 1e-1 for 10% training data and 4e-2 for 25% training data. For the classifier, we use training epoch 100, learning rate 1e-3, and train batch 32 for both ModelNet10 and ModelNet40. We report average instance accuracy for test datasets.

**Balancing of Neuron Activity** Figure 2 shows the neuron activity of each output neuron for the activity-agnostic and activity-aware learning in the same model. x-axis in the figure is the index of output neurons, and y-axis is the normalized number of the neuron activation for ModelNet10 test dataset. The activity is first summed over the all test samples and then normalized by the entire activities. In NeAW-H and NeAW-aH learning cases, the activity is dominated by few neurons, implying only few neurons are used to represent the objects. On the other hand, NeAW learning clearly displays the more distributed activity than activity-agnostic cases. In addition, the activity level increases for non-active neurons and decreases for high-active neurons, enabling more output neurons contribute to the representation. We also study how the neuron activity differs depending on the object classes. Figure 10 and 11 in Appendix C show the average neuron activity for the 10 object classes in NeAW learning and NeAW-H and NeAW-aH learning. It displays that the neurons in NeAW learning object-dependently activate, thereby, increasing the dissimilarity between the objects in the latent space. However, the neurons in the NeAW-H and NeAW-aH learning often activate regardless of the object classes. We study the correlation between the object-dependent activation and better representation learning in Appendix B. Also, the causal link between the biased activity and poor learning is studied in detail.

Table 1: Comparison with other unsupervised learning rules.

| Learning Rule | ModelNet-10 | | ModelNet-40 | |
|---|---|---|---|---|
| | Variance | Accuracy (%) | Variance | Accuracy (%) |
| Hebb's rule | 0.9663 | $72.22 \pm 0.46$ | 0.9391 | $41.52 \pm 1.21$ |
| Grossberg's rule | 0.8714 | $74.45 \pm 0.56$ | 0.8693 | $55.96 \pm 0.39$ |
| Oja's rule | 0.9182 | $80.40 \pm 0.03$ | 0.8953 | $61.22 \pm 3.82$ |
| **Ours - NeAW-H** | 0.9165 | $74.19 \pm 0.21$ | 0.9251 | $49.83 \pm 4.72$ |
| **Ours - NeAW-aH** | 0.8624 | $64.63 \pm 0.60$ | 0.8511 | $35.65 \pm 0.80$ |
| **Ours - NeAW** | 0.9793 | $88.22 \pm 0.62$ | 0.9741 | $76.20 \pm 0.30$ |

**Evolution of Neuron Activity during Unsupervised Learning**     We study the evolution of neuron activity and test accuracy as a function of training epochs of the unsupervised encoder. In all the cases, the unsupervised encoder is frozen after training with Hebbian learning for certain number of epochs, followed by full supervised training of the classifier. Figure 5 shows the variance of neuron activity during training of the model. We observe that variance is quickly saturated to a high value in NeAW Hebbian learning. In comparison, the variance increases slowly and to a lower value with NeAW-H learning and NeAW-aH learning, respectively. Figure 5(c) and (d) represent the test accuracy for ModelNet10 and ModelNet40 at different training epochs with NeAW Hebbian learning. We observe the variance and accuracy converge in 50 training epochs.

**Comparison with other Hebbian learning**     Figure 4 show t-SNE plots for the Hebb's rule, Grossberg's rule, Oja's rule, and NeAW learning. It qualitatively indicates that the NeAW learning is superior to the other Hebbian learning rules as the context vectors are well clustered with their labels. We also study the neuron activity distribution shown in Figure 2 for the other rules. The neuron activity of the NeAW learning is also well balanced than the other rules (see Figure 9 in Appendix C.) The Grossberg's rule and Oja's rule still display the skewed neuron activity dominated by the few neurons while the Hebb's rule achieves the relatively balanced activity. However, we observe that the Hebb's rule has many neurons that activate regardless of the object, indicating the less object-dependent activation than the Grossberg's rule and Oja's rule as shown in Figure 12 in Appendix C. Table 1 compares the variance of the neuron activity and test accuracy with the plain Hebb's rule, Grossberg's rule, Oja's rule, and NeAW Hebbian learning. The test accuracy is evaluated on the trained models with 5 different random seeds, and the mean and variance of the accuracy are listed in the table. We observe the NeAW Hebbian learning shows the highest variance of the neuron activity and improved test accuracy in both the datasets. In particular, the accuracy gap increases in ModelNet40, implying the potential pitfalls of the other Hebbian learning rules on complex 3D object datasets. Note, though the plain Hebb's rule show the higher variance of the neuron activity, which indicates the balanced activity, than the Oja's rule, the test accuracy is lower. This is due to the unbounded norm of the weight vectors. In other words, instead of learning the points that distinguish objects with different labels, commonly appeared points in the training samples continuously strengthens the certain weights (see Figure 20 in Appendix F.) It can increase the point-to-point variation but sacrifice local features.

**Comparison with Existing Supervised Learning Models**     Table 2 compares the proposed model with prior models for 3D object recognition. First, our model shows similar accuracy to other self-supervised model (Wu et al., 2016; Sharma et al., 2016). Accuracy difference with 3D-GAN increases in ModelNet40 but 3D-GAN needs to reconstruct a $(30, 30, 30)$ voxel to learn the representation resulting in high computational cost for high-resolution 3D objects. Our model shows similar accuracy to other fully supervised voxel-based approaches while using $6.2\times$ fewer parameters (Wu et al., 2015). The accuracy difference with other large point-based approaches is more pronounced in ModelNet40 but they use $1.8\times$ to $4.3\times$ more parameters than our model. We also compare unsupervised version and a fully supervised version (i.e., the encoder is trained with backpropagation instead of Hebbian learning) of our model. The proposed unsupervised version shows a marginally lower accuracy from the supervised version. In addition, we observe that the unsupervised learning preserves the accuracy well in smaller models than the supervised learning. Figure 16 in Appendix E shows that 0.09MB unsupervised model achieves 72.5% for ModelNet40 while the supervised

Table 2: Overall accuracy (%) and model size comparison with supervised models.

| Model | Learning | ModelNet10 | ModelNet40 | Size (MB) |
|---|---|---|---|---|
| 3DShapeNet (Wu et al., 2015) | Supervised | 83.5 | 77.0 | 19.2 |
| PointNet (Qi et al., 2017a) | Supervised | - | 89.2 | 13.2 |
| PointNet++ (Qi et al., 2017b) | Supervised | - | 91.9 | 5.6 |
| DGCNN (Wang et al., 2019) | Supervised | - | 92.9 | 6.9 |
| VConv-DAE (Sharma et al., 2016) | Self-supervised | 80.5 | 75.5 | - |
| 3D-GAN (Wu et al., 2016) | Self-supervised | 91.0 | 83.3 | - |
| **Ours - Backpropagation** | Self-supervised | 92.2 | 79.5 | 4.1 |
| **Ours - Backpropagation** | Supervised | 91.1 | 77.2 | 3.1 |
| **Ours - NeAW Hebbian** | Unsupervised | 88.2 | 76.2 | 3.1 |

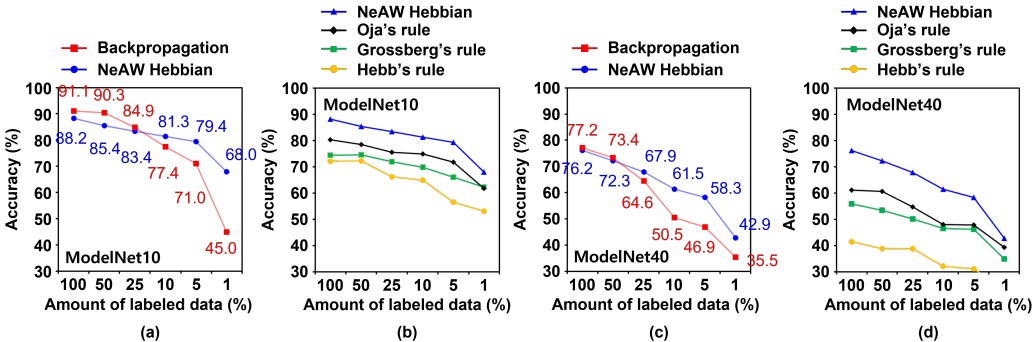

Figure 6: Comparison with other learning rules in limited training data. (a) the accuracy of end-to-end supervised learning (backpropagation) and NeAW Hebbian learning for ModelNet10 and (c) ModelNet40, (b) and the accuracy of other unsupervised learning for ModelNet10 and (d) Model-Net40.

learning in the same model shows 59.9%. Our model also includes self-supervised learning while having more parameters and computational costs (Appendix E).

**Learning with Less Training Data.** We study the performance of NeAW Hebbian as the amount of the training samples decreases (Figure 6). We randomly drop samples from each class based on the percentage amount of the labeled data. For the other Hebbian learning rules, NeAW Hebbian learning always shows higher accuracy in both ModelNet10 and ModelNet40. As shown in Table 2, NeAW Hebbian learning achieves marginally lower accuracy to fully supervised model (denoted as Backpropagation) when the all data is used for training. However, in all other cases with less than 25% labeled data, NeAW Hebbian learning shows higher performance than Backpropagation. Our model also shows better performance than alternative Hebbian rules at reduced training data.

## 5    CONCLUSION

We present the neuron activity aware (NeAW) plasticity in Hebbian learning that combines Hebbian and anti-Hebbian learning to relieve the biased neuron activity. Multi theorems and corollaries are developed to understand how NeAW Hebbian learning relieves the biased neuron activity and the correlation between the balanced activity and better representation learning in the classification task. Experimental results demonstrate that NeAW Hebbian learning achieves higher accuracy than other Hebbian learning rules and supervised learning with limited training data. We believe the paper provides a theoretical understanding of how neuron activity can be used for modulating the learning rule and its application to unsupervised 3D learning.

ACKNOWLEDGMENTS

This material is based on work sponsored by the Office of Naval Research under Grant Number N00014-20-1-2432. The views and conclusions contained in this document are those of the authors and should not be interpreted as representing the official policies, either expressed or implied, of the Office of Naval Research or the U.S. Government.

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

## A    PROOF OF THEOREM

**Theorem 1.** *Let two weight vectors $\mathbf{W}_{:,j}$ and $\mathbf{W}_{:,j'}$ with the biased activity and an input vector $\mathbf{x}_i$ satisfy $\|\mathbf{x}_i - \mathbf{W}_{:,j}(t)\|_2 < \|\mathbf{x}_i - \mathbf{W}_{:,j'}(t)\|_2$ at t. Then, NeAW Hebbian learning changes the sign of the inequality to $\|\mathbf{x}_i - \mathbf{W}_{:,j}(t+1)\|_2 > \|\mathbf{x}_i - \mathbf{W}_{:,j'}(t+1)\|_2$ at $t+1$ if the vectors are on the geometric condition $(1 + \frac{\eta}{N})\|\mathbf{x}_i - \mathbf{W}_{:,j}(t)\|_2 > (1 - \frac{\eta}{N})\|\mathbf{x}_i - \mathbf{W}_{:,j'}(t)\|_2$.*

*Proof.* Let's say there are an input vector $\mathbf{x}_i$, corresponding winner is the $j$-th output neuron (i.e. $j = \arg\min_k \|\mathbf{x}_i - \mathbf{W}_{:,k}(t)\|$), and the other loser is the $j'$-th output neuron. We are interested in how the hybrid update changes the activation boundary. Our learning rule is given by:

$$\mathbf{W}_{:,j}(t+1) = \mathbf{W}_{:,j}(t) + f(p^* - p^{(j)})\frac{\eta}{N} \sum_{i=0}^{N-1} \mathbf{1}_{i \in \arg\min_k \|\mathbf{x}_k - \mathbf{W}_{:,j}(t)\|_2}(\mathbf{x}_i - \mathbf{W}_{:,j}(t)) \quad (7)$$

Given there is a single input vector, the learning rule can be simplified by:

$$\mathbf{W}_{:,j}(t+1) = \mathbf{W}_{:,j}(t) + f(p^* - p^{(j)})\frac{\eta}{N}(\mathbf{x}_i - \mathbf{W}_{:,j}(t)) \quad (8)$$

As we are interested in the hybrid update relieves the biased activity such as too high activity or zero activity, the $j$-th and $j'$-th neuron are assumed to have higher $p^{(j)}$ and lower $p^{(j')}$ than $p^*$. Then, with an assumption that $a$ and $b$ are 1 to simplify the case, the equation is again written as:

$$\mathbf{W}_{:,j}(t+1) = \mathbf{W}_{:,j}(t) - \frac{\eta}{N}(\mathbf{x}_i - \mathbf{W}_{:,j}(t)) \quad (9)$$

$$\mathbf{W}_{:,j'}(t+1) = \mathbf{W}_{:,j'}(t) + \frac{\eta}{N}(\mathbf{x}_i - \mathbf{W}_{:,j'}(t)) \quad (10)$$

Our goal is to show that the learning rules change the activation boundary. At $t$ before learning, the Euclidean distance for the $j$-th neuron is lower than the $j'$-th neuron, or more formally:

$$\|\mathbf{x}_i - \mathbf{W}_{:,j}(t)\|_2^2 < \|\mathbf{x}_i - \mathbf{W}_{:,j'}(t)\|_2^2 \quad (11)$$

$$\|\mathbf{x}_i - \mathbf{W}_{:,j}(t)\|_2^2 - \|\mathbf{x}_i - \mathbf{W}_{:,j'}(t)\|_2^2 < 0 \quad (12)$$

$$(\mathbf{W}_{:,j}(t) + \mathbf{W}_{:,j'}(t) - 2\mathbf{x}_i) \cdot (\mathbf{W}_{:,j}(t) - \mathbf{W}_{:,j'}(t)) < 0 \quad (13)$$

At $t+1$ after learning, the left-hand side of (13) is written as:

$$(\mathbf{W}_{:,j}(t+1) + \mathbf{W}_{:,j'}(t+1) - 2\mathbf{x}_i) \cdot (\mathbf{W}_{:,j}(t+1) - \mathbf{W}_{:,j'}(t+1)) \quad (14)$$

The hybrid update changes the winner neuron if it is positive. Using (9) and (10), (14) can be re-written as:

$$\begin{aligned}
&\{\mathbf{W}_{:,j}(t) + \mathbf{W}_{:,j'}(t) - 2\mathbf{x}_i + \frac{\eta}{N}(\mathbf{W}_{:,j}(t) - \mathbf{W}_{:,j'}(t))\} \cdot \\
&\{\mathbf{W}_{:,j}(t) - \mathbf{W}_{:,j'}(t) - \frac{\eta}{N}(2\mathbf{x}_i - \mathbf{W}_{:,j}(t) - \mathbf{W}_{:,j'}(t))\}
\end{aligned} \quad (15)$$

$$\frac{4\eta}{N}\|\mathbf{x}_i - \mathbf{W}_{:,j}(t)\|_2^2 + (1 - \frac{\eta}{N})^2(\mathbf{W}_{:,j}(t) + \mathbf{W}_{:,j'}(t) - 2\mathbf{x}_i) \cdot (\mathbf{W}_{:,j}(t) - \mathbf{W}_{:,j'}(t)) \quad (16)$$

Substituting the second term in (16) by the left-hand side of (12),

$$(1 + \frac{\eta}{N})^2 \|\mathbf{x}_i - \mathbf{W}_{:,j}(t)\|_2^2 - (1 - \frac{\eta}{N})^2 \|\mathbf{x}_i - \mathbf{W}_{:,j'}(t)\|_2^2 \tag{17}$$

Equation (14) is positive if

$$\frac{\|\mathbf{x}_i - \mathbf{W}_{:,j'}(t)\|_2^2}{\|\mathbf{x}_i - \mathbf{W}_{:,j}(t)\|_2^2} < \frac{(1 + \frac{\eta}{N})^2}{(1 - \frac{\eta}{N})^2} \tag{18}$$

∎

**Corollary 1.1.** *Let two weight vectors* $\mathbf{W}_{:,j}$ *and* $\mathbf{W}_{:,j'}$ *with the biased activity and an input vector* $\mathbf{x}_i$ *satisfy* $\|\mathbf{x}_i - \mathbf{W}_{:,j}(t)\|_2 < \|\mathbf{x}_i - \mathbf{W}_{:,j'}(t)\|_2$ *at* $t$. *Then, Hebbian learning (NeAW-H) preserves the sign of the inequality to* $\|\mathbf{x}_i - \mathbf{W}_{:,j}(t+1)\|_2 < \|\mathbf{x}_i - \mathbf{W}_{:,j'}(t+1)\|_2$ *at* $t + 1$.

*Proof.* Similar to the proof of Theorem 1, we need to show the equation below is positive if Hebbian learning changes the winner neuron:

$$(\mathbf{W}_{:,j}(t+1) + \mathbf{W}_{:,j'}(t+1) - 2\mathbf{x}_i) \cdot (\mathbf{W}_{:,j}(t+1) - \mathbf{W}_{:,j'}(t+1)) \tag{19}$$

We have Hebbian learning rules for both the $j$-th and $j'$-th neuron as:

$$\mathbf{W}_{:,j}(t+1) = \mathbf{W}_{:,j}(t) + \frac{\eta}{N}(\mathbf{x}_i - \mathbf{W}_{:,j}(t)) \tag{20}$$

$$\mathbf{W}_{:,j'}(t+1) = \mathbf{W}_{:,j'}(t) + \frac{\eta}{N}(\mathbf{x}_i - \mathbf{W}_{:,j'}(t)) \tag{21}$$

Using the learning rules to re-write (19),

$$\{\mathbf{W}_{:,j}(t) + \mathbf{W}_{:,j'}(t) - 2\mathbf{x}_i + \frac{\eta}{N}(2\mathbf{x}_i - \mathbf{W}_{:,j}(t) - \mathbf{W}_{:,j'}(t))\} \cdot$$
$$\{\mathbf{W}_{:,j}(t) - \mathbf{W}_{:,j'}(t) - \frac{\eta}{N}(\mathbf{W}_{:,j}(t) - \mathbf{W}_{:,j'}(t))\} \tag{22}$$

$$(1 - \frac{\eta}{N})^2 (\mathbf{W}_{:,j}(t) + \mathbf{W}_{:,j'}(t) - 2\mathbf{x}_i) \cdot (\mathbf{W}_{:,j}(t) - \mathbf{W}_{:,j'}(t)) \tag{23}$$

As we assume

$$(\mathbf{W}_{:,j}(t) + \mathbf{W}_{:,j'}(t) - 2\mathbf{x}_i) \cdot (\mathbf{W}_{:,j}(t) - \mathbf{W}_{:,j'}(t)) < 0 \tag{24}$$

Equation (19) is always negative. Thus Hebbian learning cannot change the winner neuron. ∎

**Corollary 1.2.** *Let two weight vectors* $\mathbf{W}_{:,j}$ *and* $\mathbf{W}_{:,j'}$ *with the biased activity and an input vector* $\mathbf{x}_i$ *satisfy* $\|\mathbf{x}_i - \mathbf{W}_{:,j}(t)\|_2 < \|\mathbf{x}_i - \mathbf{W}_{:,j'}(t)\|_2$ *at* $t$. *Then, anti-Hebbian learning (NeAW-aH) preserves the sign of the inequality to* $\|\mathbf{x}_i - \mathbf{W}_{:,j}(t+1)\|_2 < \|\mathbf{x}_i - \mathbf{W}_{:,j'}(t+1)\|_2$ *at* $t + 1$.

*Proof.* Similar to the proof of Theorem 1, we need to show the equation below is positive if anti-Hebbian learning changes the winner neuron:

$$(\mathbf{W}_{:,j}(t+1) + \mathbf{W}_{:,j'}(t+1) - 2\mathbf{x}_i) \cdot (\mathbf{W}_{:,j}(t+1) - \mathbf{W}_{:,j'}(t+1)) \tag{25}$$

We have anti-Hebbian learning rules for both the $j$-th and $j'$-th neuron as:

$$\mathbf{W}_{:,j}(t+1) = \mathbf{W}_{:,j}(t) - \frac{\eta}{N}(\mathbf{x}_i - \mathbf{W}_{:,j}(t)) \tag{26}$$

$$\mathbf{W}_{:,j'}(t+1) = \mathbf{W}_{:,j'}(t) - \frac{\eta}{N}(\mathbf{x}_i - \mathbf{W}_{:,j'}(t)) \tag{27}$$

Using the learning rules to re-write (25),

$$\{\mathbf{W}_{:,j}(t) + \mathbf{W}_{:,j'}(t) - 2\mathbf{x}_i + \frac{\eta}{N}(2\mathbf{x}_i - \mathbf{W}_{:,j}(t) - \mathbf{W}_{:,j'}(t))\} \cdot$$
$$\{\mathbf{W}_{:,j}(t) - \mathbf{W}_{:,j'}(t) - \frac{\eta}{N}(\mathbf{W}_{:,j}(t) - \mathbf{W}_{:,j'}(t))\} \tag{28}$$

$$(1 - \frac{\eta}{N})^2(\mathbf{W}_{:,j}(t) + \mathbf{W}_{:,j'}(t) - 2\mathbf{x}_i) \cdot (\mathbf{W}_{:,j}(t) - \mathbf{W}_{:,j'}(t)) \tag{29}$$

As we assume
$$(\mathbf{W}_{:,j}(t) + \mathbf{W}_{:,j'}(t) - 2\mathbf{x}_i) \cdot (\mathbf{W}_{:,j}(t) - \mathbf{W}_{:,j'}(t)) < 0 \tag{30}$$

Equation (25) is always negative. Thus anti-Hebbian learning cannot change the winner neuron. ∎

**Theorem 2.** *For a network of $N$ neurons in the biased activity with weight vectors $\mathbf{W}$ and an input vector $\mathbf{x}$, the variance of the distribution of activated neurons follows* $\mathrm{Var}(\text{NeAW Hebbian}) \geq \mathrm{Var}(\text{Hebbian}) \, \text{or} \, \mathrm{Var}(\text{anti} - \text{Hebbian})$ *which indicates NeAW Hebbian learning provides the greatest variance of the neuron activity compared to solely using Hebbian or anti-Hebbian learning.*

*Proof.* First, we note the results shown by Plumbley (1993), that Hebbian/anti-Hebbian networks optimize the information capacity by orthonormalizing the principal subspace. In order to prove that the neuronal activations for the hybrid learning rule have a greater variance than the normal Hebbian or anti-Hebbian learning rules, we first assume that a network with a greater number of neuronal activations can encode a greater number of principal components in the subspace. We consider that the local Hebbian and anti-Hebbian learning rules recover a principal subspace from the data using a principled cost function as shown by Pehlevan et al. (2015).

The subspace method of classification is pattern recognition method where the primary model for a class is a linear subspace of the Euclidean pattern space (Watanabe & Pakvasa, 1973). Let $u_i \in \mathbb{R}^n, i = 1 \dots m, m < n$ be a set of $m$ linearly independent vectors that spans the subspace $L$ as:

$$P = P(u_1, \dots, u_m) = \left\{ x \mid x = \sum_{i=1}^{m} \alpha_i u_i \right\}$$

where $\alpha_i$ is some scalar. That is, essentially, the problem can be re-written as:

$$\mathrm{Var}(\text{NeAW Hebbian}) \geq \mathrm{Var}(\text{Hebbian}) \, \text{or} \, \mathrm{Var}(\text{anti} - \text{Hebbian})$$
$$\Rightarrow \mathrm{span}_{\min}(\mathcal{L}_{\text{NeAW Hebbian}}) \geq \mathrm{span}_{\min}(\mathcal{L}_{\text{Hebbian}}) \, \text{or} \, \mathrm{span}_{\min}(\mathcal{L}_{\text{anti-Hebbian}}) \tag{31}$$

where $\mathrm{span}_{\min}$ is the minimum spanning subspace, i.e., the subspace does not have a smaller subspace spanned by a subset of the samples. For a detailed explanation of minimal sample subspace refer to the works of Zhang & Xia (2019).

As shown before, the learning rules for the Hebbian and anti-Hebbian cases are given, respectively as:

$$\mathbf{W}_{:,j}^{\text{Hebb}}(t+1) = \mathbf{W}_{:,j}^{\text{Hebb}}(t) + \frac{\eta}{N}(\mathbf{x}_i - \mathbf{W}_{:,j}(t))$$
$$\mathbf{W}_{:,j}^{\text{anti-Hebb}}(t+1) = \mathbf{W}_{:,j}^{\text{anti-Hebb}}(t) - \frac{\eta}{N}(\mathbf{x}_i - \mathbf{W}_{:,j}(t)) \tag{32}$$

Now, let us consider the case where we have $N$ neurons which are learned using Hebbian or anti-Hebbian learning rules. We aim to bifurcate the sets of trained neurons into Hebbian and anti-Hebbian classes. Without loss of generality, we can assume that the first $n$ ($n < N$) neurons are trained using the a learning method $\mathcal{L} \in \{\mathcal{L}_H = \text{Hebbian}, \mathcal{L}_{aH} = \text{anti-Hebbian}\}$. Mathematically, we do a binary partition of the metric space of the neurons and the weights. If we consider the metric space of the neuron vectors and the weight vectors to be represented as $\mathcal{W}$, we define the partition from $X \rightarrow Y, Z$ where $X$ denotes the space of all the vectors, $Y$ denotes the space of vectors where all the neurons have already learnt (which is interpreted as a coloring of the vector) using learning algorithm $\mathcal{L}$. $Z$ denotes the space of vectors where none of the neurons have been assigned a learning rule, i.e., they can learn using either $\mathcal{L}_H$ or $\mathcal{L}_{aH}$. We denote the span of the subspace learned using the learning method $\mathcal{L}$ is denoted as $\text{span}(x_1^{\mathcal{L}}, \ldots, x_n^{\mathcal{L}})$.

Now, let us consider the $n+1$-th neuron. There might be two cases that might arise: (i) the neuron is trained using the same learning rule $\mathcal{L}$ with which the previous $n$ neurons are trained. (ii) the neuron is trained using the complementary learning rule $\bar{\mathcal{L}}$. Thus, we study the differential gain by using one learning rule over the other for the $n + 1$-th neuron.

We know that a learning rule basically extracts the principal components of the input space by projecting the input space into the orthogonal subspace. Also, we know that a neural network with a richer subspace, i.e., a subspace with a larger number of orthogonal principal components, can learn a better representation of the input space than a subspace with a fewer number of orthogonal principal components.

Again, let us consider the subspaces learned by the Hebbian $\mathcal{L}_H$ and the anti-Hebbian $\mathcal{L}_{aH}$ learning rules. From previous works (Rubner & Tavan, 1989; Carlson, 1990; Schraudolph & Sejnowski, 1991) we know that anti-Hebbian and Hebbian learning rules lead to different principal components. As described by Schraudolph & Sejnowski (1991) while the built-in temporal smoothness constraint enables Hebbian neurons to learn the invariance classes, anti-Hebbian synapses have been used for lateral decorrelation of feature detectors and removal of temporal variations from the input. A set of Hebbian feature detectors whose weight vectors span the hyperplane would characterize the associated class of stimuli. The anti-Hebbian learning algorithm, however, provides a more efficient representation when the dimensionality of the hyperplane is more than half of the input space. This is because, in that case, we require fewer normal vectors than spanning vectors for unique characterization. Since anti-Hebbian neurons remove the variance within a stimulus class, they present a different output representation to subsequent layers than the Hebbian neurons. Thus, since the linear subspaces learned by the two learning rules are distinct, we can say that:

$$\text{span}(x_1^{\mathcal{L}}, \ldots, x_n^{\mathcal{L}} \cup x_{n+1}^{\mathcal{L}}) \subseteq \text{span}(x_1^{\mathcal{L}}, \ldots, x_n^{\mathcal{L}} \cup x_{n+1}^{\bar{\mathcal{L}}})$$
$$\Rightarrow \text{span}(\text{Hebbian}) \text{ or span}(\text{anti-Hebbian}) \subseteq \text{span}(\text{NeAW Hebbian}) \tag{33}$$

Thus, we see that using a hybrid Hebbian learning rule increases the span of the subspace learned, i.e., increases the variance of the neuron activations. ∎

**Corollary 2.1.** *Hebbian or anti-Hebbian learning on the biased neuron activity leads to a poor subspace representation than NeAW Hebbian learning*

*Proof.* As we showed before in Theorem 2 in (33), we see that the span of the subspace learned by the NeAW Hebbian learning algorithm is greater than the span of the homogeneous Hebbian or anti-Hebbian learning method. Let us label each neuron as 'H' or 'aH' for Hebbian or anti-Hebbian learning methods depending on the learning algorithm it selects. Now, let us denote the distribution of the nodes marked 'H' and the distribution of the nodes marked 'aH' in the vector space of the neuron weights. We notice that since the total number of neurons is fixed, the total probability mass of these two distributions is also fixed. Thus, if there are a lot more 'H' neurons, the probability mass of the 'aH' neurons is lower - i.e., for the set of neurons $n_i \in \mathcal{N}$, $\mathbb{P}[\sum_{i \in H \sim \mathcal{N}} n_i] > \mathbb{P}[\sum_{i \in aH \sim \mathcal{N}} n_i]$. Now, considering the full set of principal components that the entire network can learn using an infinite number of neurons. As we keep decreasing the number of neurons, we keep learning a subset of this ideal set of principal components. Since the sum of probability masses of the Hebbian and anti-Hebbian learning algorithms are fixed, if the distribution is heavily biased

towards just Hebbian learning, then the principal axes that could be learned from the anti-Hebbian learning are not present. Thus, if we represent the principal axes of the learning rule $\mathcal{L}$ as $\mathbf{a}_{\mathcal{L}}$, then $\text{span}_{\min}(\sum \mathbf{a}_X) \subseteq \text{span}_{\min}(\sum \mathbf{a}_{\text{NeAW}})$, where $X$ represents either Hebbian or anti-Hebbian learning neurons. Hence, a biased learning algorithm leads to a poorer selection of the principal axes, which leads to worse performance. ∎

## B  CAUSAL RELATION BETWEEN NEURON ACTIVITY AND REPRESENTATION LEARNING

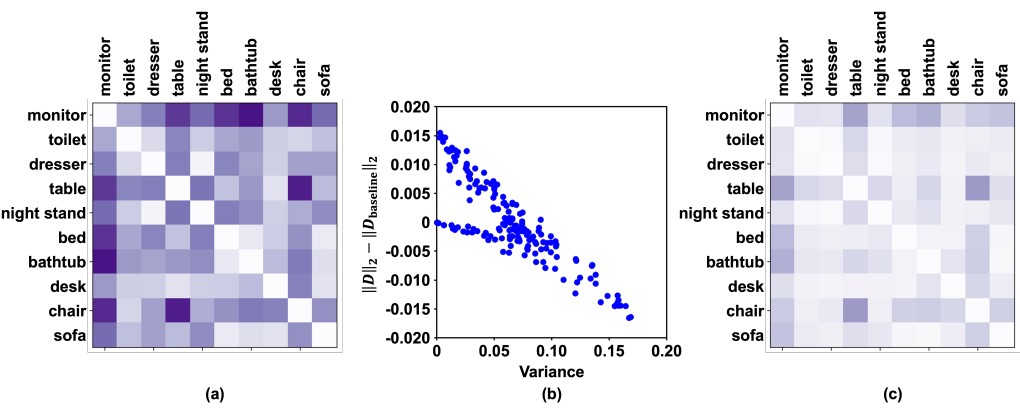

(a)       (b)       (c)

Figure 7: (a) Dissimilarity matrix for ModelNet10 after NeAW learning, (b) L2 norm change of the dissimilarity matrix by deactivating a neuron in NeAW leanring, and (c) dissimilarity matrix after learning using Hebb's rule. The x-axis in (b) indicates the variance of the deactivated neuron's activity for different object classes, and the y-axis is the L2 norm change of the dissimilarity matrix by the deactivation.

**Causal Relation between the Biased Activity and Poor Learning**  Hebbian-like unsupervised learning rules with Winner-Take-All (WTA) can be interpreted as a clustering module. We focus on the factors that affect the quality of clustering to build a causal connection between the neuron activity and the performance. It is well-known that (1) the number of cluster centroids and (2) their separability play a significant role in determining the quality of clustering (Kodinariya & Makwana, 2013). If there are too few cluster centroids, irrelevant points will be associated with the same cluster centroid. Likewise, if the cluster centroids are not separable, i.e., multiple centroids are positioned at the same location, the effective number of clusters will be still low. In the end, we need the *enough number of well separable cluster centroids* that behave as principal components for different local features in the object. In Corollary 2.1 (Appendix A), we have proved that Hebbian and anti-Hebbian learning leads to biased neuron activity and hence, fewer principal components than NeAW learning. Fewer principal components indicate fewer separable cluster centroids which leads to poor performance. Hence, using Corollary 2.1 we can state that a biased activity in the Hebbian and anti-Hebbian learning causes a poor performance. That is, the biased activity problem should be resolved to improve the performance. However, it does not necessarily indicate that the balanced activity leads to a better performance.

**Correlation between the Balanced Activity and Better Clustering**  The causal link between biased activity and poor performance suggests that reducing activity bias may improve performance. However, it does not necessarily indicate that the only balancing activity *causes* a better performance. In particular, we observe that the "balanced activity with object-dependent activation" is correlated with the clustering quality. The idea behind is that the neurons that activate regardless of the object classes (no object dependency) will not capture the features to design the classification boundary. In this light, we design an additional experiment to quantify the change of the clustering quality depending on how dependent neurons are to different object classes. We quantify the quality using the dissimilarity matrix. The dissimilarity matrix for ModelNet10 is given by 10x10 matrix, where each element is the dissimilarity between the encoded vectors of the corresponding two object

classes (see Figure 7(a).) We choose cosine similarity to calculate the dissimilarity. Let $x_A$ and $x_B$ for the encoded vector for A and B objects. Then, the dissimilarity in the matrix ($D$) is given by $D[A][B] = 1 - \frac{x_A \cdot x_B}{|x_A||x_B|}$. We calculate the change of the L2 norm of the dissimilarity matrix by deactivating neurons one by one. The experiment results demonstrate that deactivating the neuron that object-dependently activates reduces the dissimilarity between the objects, indicating a poor clustering quality. In contrast, deactivating the neuron that activates regardless of the object labels increases the dissimilarity, leading to a better clustering quality (see Figure 7(b).) Note, the number of activating neurons during the experiment are same as we deactivate a single neuron at a time.

Figure 7(c) shows that the dissimilarity matrix in Hebb's rule, and all the matrix elements in NeAW are higher than Hebb's rule. It indicates that the object classes are easily distinguishable in NeAW. As many neurons activate regardless of the objects in Hebb's rule, the inner product between the encoded vectors increases, and the cosine-based dissimilarity decreases leading to the poor clustering quality. That is, the encoded vectors with less common activation will have the higher dissimilarity, which indicates the better clustering quality. In the end, the higher dissimilarity is preferred; and we consider the L2 norm of the dissimilarity matrix to compare the cluster quality between the learning rules. The L2 norm for Hebb's rule, Grossberg's rule, Oja's rule, and NeAW learning are 0.630, 1.049, 1.313, and 1.482, respectively. Note, the accuracy of them for ModelNet10 are 72.22%, 74.45%, 80.40%, and 88.22%, respectively.

In summary, the biased activity causes a poor learning in the proposed model while the balanced activity is not a cause of the better performance. The additional experiment demonstrates that the clustering quality varies depending on both the activity balance and the neuron's object dependency.

## C  NEURON ACTIVITY DISTRIBUTION IN OTHER LEARNING RULE

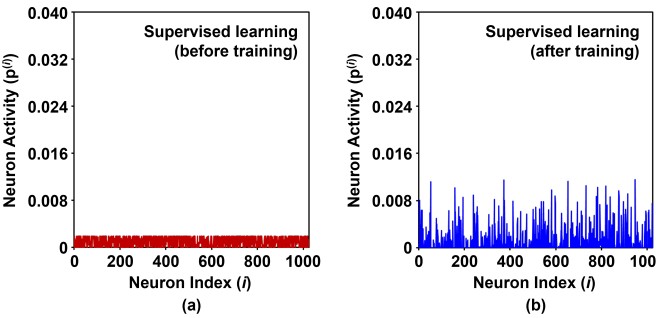

Figure 8: The average neuron activity of each output neuron for ModelNet10 (a) before and (b) after end-to-end supervised learning.

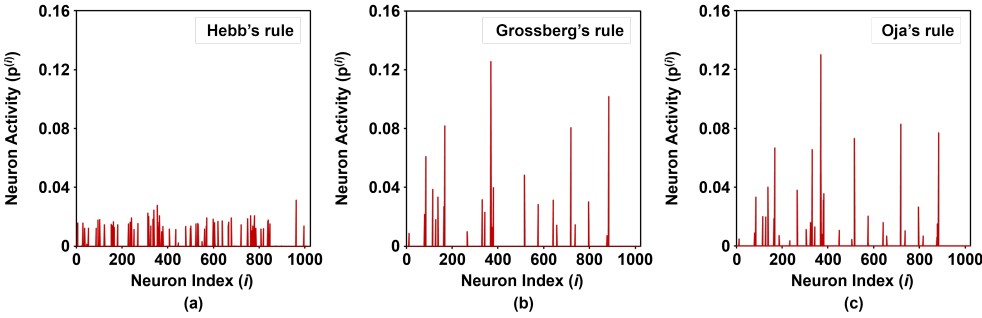

Figure 9: The average neuron activity of each output neuron for ModelNet10 with (a) Hebb's rule, (b) Grossberg's rule, and (c) Oja's rule.

We study the neuron activity distribution of the supervised model and the other Hebbian learning rules. The neuron activity here is the average number of the activation calculated by the same way in Figure 2.

**Neuron Activity in Supervised Learning**    Note, our end-to-end supervised model does not have the WTA modules as they are not differentiable while the other Hebbian learning rules are trained with the WTA modules. We observe that the neuron activity of the supervised model is evenly distributed before training as shown in Figure 8(a). This will be related to the random initialization of the weights, which randomly activate the neuron through ReLU function. Figure 8(b) displays that the activity is less balanced after training in the supervised model though there is no highly skewed activity observed in the Hebbian and anti-Hebbian learning.

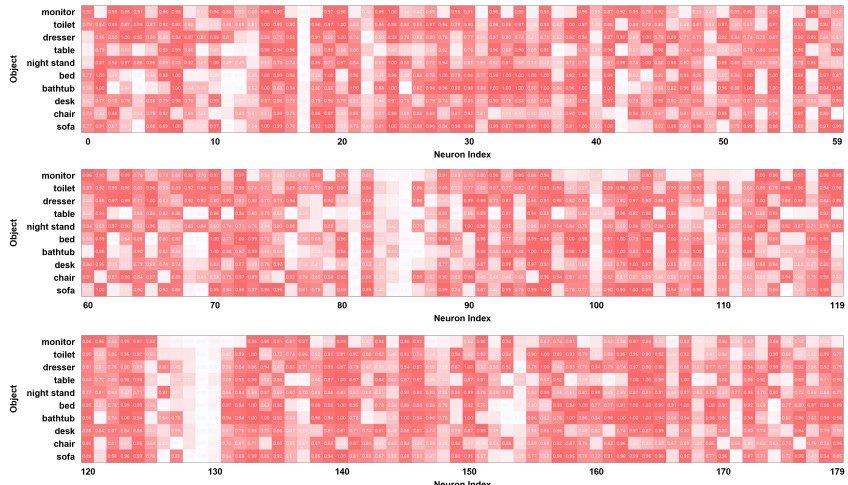

Figure 10: Neuron activity for different object class in NeAW learning. 180 neurons are displayed among total 1024 output neurons as the others do not activate for all the test samples.

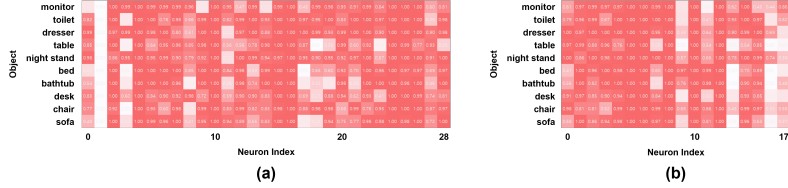

Figure 11: Neuron activity for different object class in (a) NeAW-H learning and (b) NeAW-aH learning. 29 and 18 neurons for Hebbian and anti-Hebbian learning, respectively, are displayed among total 1024 output neurons as the others do not activate for all the test samples.

**Neuron Activity in other Hebbian Learning**    We perform the experiment for the other Hebbian learning rules such as Hebb's rule, Grossberg's rule, and Oja's rule. Figure 9 shows the average neuron activity is also biased in Grossberg's rule and Oja's rule, which is similar with NeAW-H and NeAW-aH learning in Figure 2. Hebb's rule has the relatively balanced activity than the others. However, it does not lead to the better accuracy as summarized in Table 1. We qualitatively observe that all the learning rules have the less balanced activity than the NeAW learning. It parallels with the variance of the neuron activity in Table 1.

**Object-wise Neuron Activity in Hebbian Learning Rules**    Figure 2, 8, and 9 show the average neuron activity for all the test samples. However, the average neuron activity for the different object class is not presented. First, Figure 10 and 11 display the average neuron activity in NeAW learning and NeAW-H and NeAW-aH learning, respectively. Each column in the figure represents a neuron index, and row indicates an object label. The value of elements is normalized, where the maximum

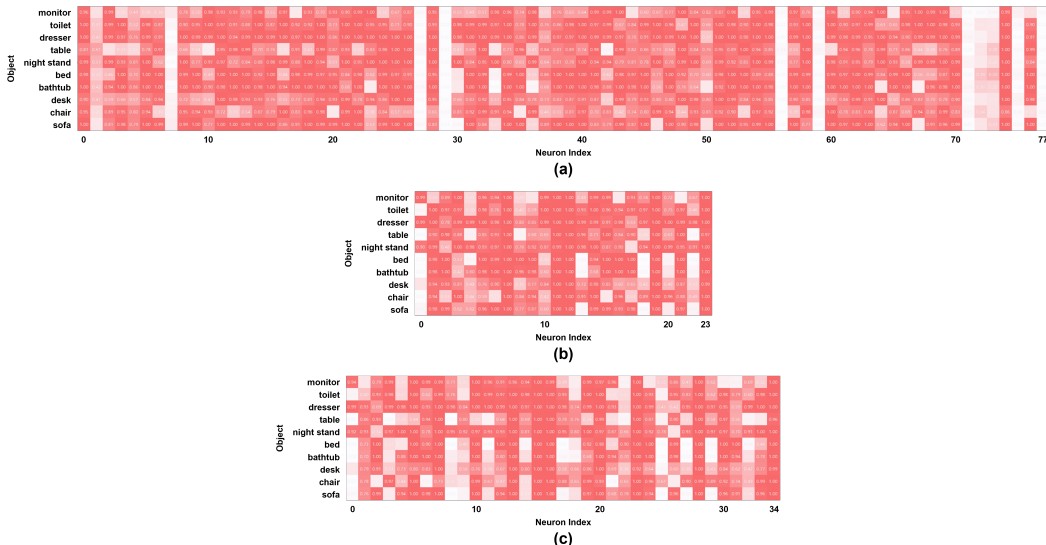

Figure 12: The neuron activity for different object class in (a) Hebb's rule, (b) Grossberg's rule, and (c) Oja's rule.

value is 1 (red) indicating the neuron activates for all the object samples while the minimum value is 0 (white) meaning the neuron never activates for this object. The figures represent that the number of activating neurons is higher in the NeAW learning than the activity-agnostic NeAW-H and NeAW-aH learning, and more importantly, the neurons selectively activate depending on the object labels. Similarly, Figure 12 shows the average activation in Hebb's rule, Grossberg's rule, and Oja's rule. As we view the neuron activation in WTA modules as the cluster centroid, it means Hebb's rule creates many cluster centroids that group all the objects regardless of their labels, thereby, decreasing the dissimilarity between the objects in the latent space. In other words, it is hard to distinguish the rows (i.e. encoded vectors) in Figure 12(a). This is the reason why Hebb's rule gives a poor performance while having the balanced activity (Figure 9). Though Grossberg's rule and Oja's rule in Figure 12(b) and (c) display sparse columns and more object-dependent activation, the number of activating neurons is small (∼35). In this case, they create separable cluster centroids but the number of them are too few.

## D  IMPACT OF HYPERPARAMETERS IN NEAW LEARNING

We evaluate the impact of hyperparameters $a$ and $b$ in (3) with the variance of the neuron activity. Figure 13(a) and (c) show that the higher $a$ and $b$ results in a faster increase of the variance during training. This can be also understood through the geometric condition given in Theorem 1:

$$\frac{\|\mathbf{x}_i - \mathbf{W}_{:,j'}(t)\|_2^2}{\|\mathbf{x}_i - \mathbf{W}_{:,j}(t)\|_2^2} < \frac{(1 + \frac{\eta}{N})^2}{(1 - \frac{\eta}{N})^2} \tag{34}$$

which describes that NeAW Hebbian learning relieves the biased activity only if the ratio of the Euclidean distance are lower than the ratio of $\left(1 + \frac{\eta}{N}\right)$ and $\left(1 - \frac{\eta}{N}\right)$. Given $a$ and $b$ can be regarded as different learning rates $\eta$, higher $a$ and higher $b$ will increase the numerator and decrease the denominator, respectively. It results in the geometric condition for relieving the biased activity to become weaker, and consequently, such pair of $a$ and $b$ quickly increases the variance. However, in Figure 13(b) and (d), we observe the maximum accuracy in different $a$ and $b$ values are saturated to the similar level, which implies that the final accuracy with the NeAW Hebbian learning is not sensitive to $a$ and $b$.

a=0 or b=0 cases indicate that the either Hebbian or anti-Hebbian learning is used rather than the hybridization. In the very low rates such as 0, 0.01, 0.05, and 0.1 for a and b, the saturate accuracy and variance is still at the similar level. However, it is important to note that the variance of the neuron activity slowly increases in the lower rate as shown in Figure 13(a) and (c), indicating the

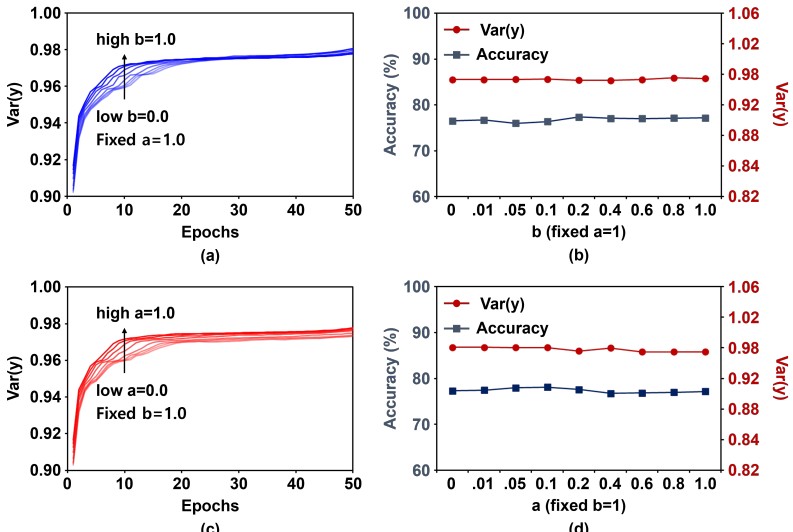

Figure 13: Effect of a and b for ModelNet40. (a) the variance of the neuron activity with fixed a and variable b, (b) the accuracy with the variable b values, (c) the variance of the neuron activity with variable a and fixed b, and (d) the accuracy with the variable a values.

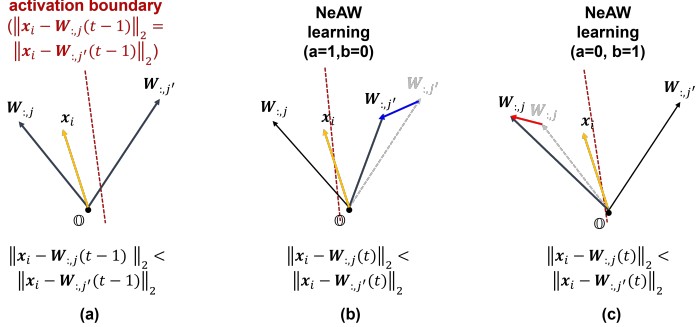

Figure 14: Geometry of input and weight vectors with different a and b values. (a) the initial geometry of the vectors and the changed geometry of the vectors after (b) NeAW learning with a=1 and b=0 and (c) NeAW learning with a=0 and b=1.

slow relaxation of the biased activity. It can be intuitively understood by the vector schematic in Figure 14. Though the NeAW learning with either a=0 or b=0 will not guarantee the fast relaxation of the biased activity, the direction of the weight update can be still properly designed. Figure 15 shows the test accuracy for ModelNet40 with the models which encoder is only trained by 5 epochs. We observe that having higher a and b generally achieves higher accuracy than having very low a and b values.

# E    PERFORMANCE ANALYSIS IN VARIOUS MODEL ARCHITECTURE

**Relation between Model Complexity and Performance**    We explore the smaller and larger model architectures with the supervised and unsupervised learning to understand how the performance changes depending on the model complexity. We design few simpler models with single-layer encoder and classifier. Figure 16 shows how the test accuracy for ModelNet40 changes in the various models using end-to-end backpropagation and NeAW Hebbian learning. We observe the accuracy of the simpler supervised models clearly drops to 46.6%∼61.9% while the unsupervised models perform better showing 56.1%∼72.5% accuracy. The performance difference is particularly large at the model which size is 0.09MB: the unsupervised model achieves 71.7% and the supervised model

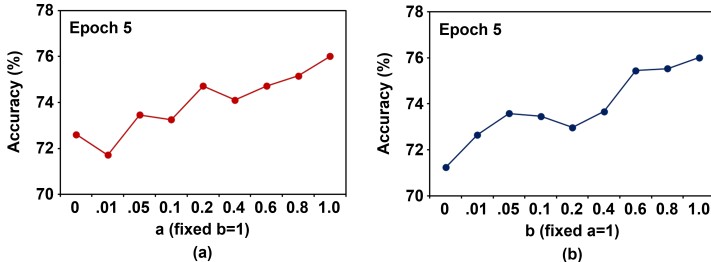

Figure 15: (a) The accuracy for ModelNet40 at the encoder training epoch 5 with the variable a values and (b) the variable b values.

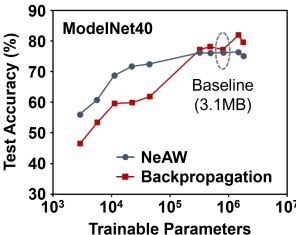

Figure 16: Test accuracy for ModelNet40 in smaller and larger models using supervised (Backprop-agation) and NeAW Hebbian learning.

shows 59.9%. For the larger models, we increase the dimension of the last layer in the encoder. However, the supervised models show slightly higher accuracy than the baseline model while the performance of the unsupervised models saturate.

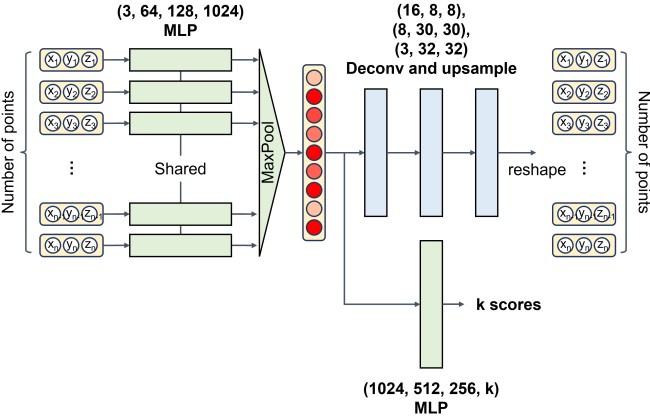

Figure 17: Self-supervised model architecture. A reconstruction decoder with deconvolution and upsample layers is added on the proposed supervised and unsupervised model architecture.

**Self-supervised Model and its Performance** We design a self-supervised learning model based on the proposed model in Table 2. Figure 17 shows the schematic of the model architecture. The self-supervised learning cannot be directly applied on the same architecture and requires another autoencoder-based decoder to reconstruct the input point cloud. Hence, a simple decoder with three deconvolution and upsample layers is added in the proposed architecture. The self-supervised model is first trained with the encoder and reconstruction decoder to learn the latent representation. The main difference in training the self-supervised model is a loss function. We use Chamfer distance loss to compare the distance between ground truth and reconstructed point cloud. After the training, the parameters of the encoder are fixed, and the classifier decoder is trained by supervised learn-

ing. We observe the self-supervised model achieves the similar accuracy in both ModelNet10 and ModelNet40 datasets with the supervised and unsupervised models. The accuracy is 92.2% for ModelNet10 and 79.5% for ModelNet40. However, we would like to note the computational cost of the reconstruction decoder. While the encoder and classifier used in the inference are same with the supervised and unsupervised models (3.1MB), the number of parameters in the simple reconstruction decoder is 263,539 (1.0MB, total 4.1MB). Also, during the training, the activation size per sample increases from 808 to 12,104 due to the reconstruction. In other words, the computational cost for the inference is same, but the overhead of training is larger in the self-supervised learning.

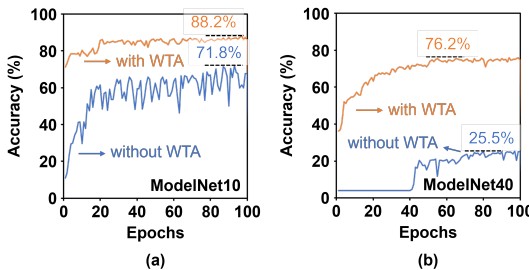

Figure 18: Test accuracy for (a) ModelNet10 and (b) ModelNet40 in NeAW Hebbian learning with and without WTA modules.

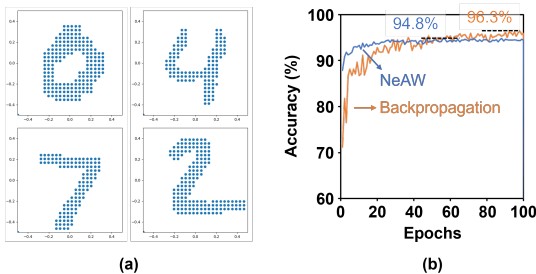

Figure 19: (a) Examples of point MNIST datasets and (b) test accuracy for point MNIST in supervised learning (Backpropagation) and NeAW Hebbian learning.

**Model Performance without WTA modules**    We would like to note that the learning rule in Equation (1) includes the indicator function. It represents that the weight update is based on the closest input point that is another WTA operation to couple with the WTA modules. Hence, the proposed learning rule is designed along with the WTA modules, which is an important contribution in the paper. Figure 18 shows the test accuracy comparison between the model with and without the WTA modules in both ModelNet10 and ModelNet40. The model without the WTA modules achieves 71.8% for ModelNet10 and 25.5% for ModelNet40 while having the WTA modules shows 88.2% for ModelNet10 and 76.2% for ModelNet40. The results show that the model performance significantly drops without the WTA modules. Thus, this ablation study demonstrate that the NeAW learning should accompany the WTA modules.

**Model Performance in Point MNIST**    We train and evaluate our supervised and unsupervised models on a simple 2D classification task using point MNIST datasets. The model architecture is same with the other experiments. Each data samples include the point cloud representation of the MNIST images where the coordinates of the dark pixels are included. Figure 19 shows that the end-to-end supervised learning achieves 96.3%, and the NeAW learning shows the 94.8%. As the datasets are relatively simple than ModelNet10 and ModelNet40, both the learning methods shows the similarly high accuracy. However, we observe that Hebb's rule, Grossberg's rule, and Oja's rule achieve 58.7%, 55.2%, and 69.0% test accuracy, respectively.

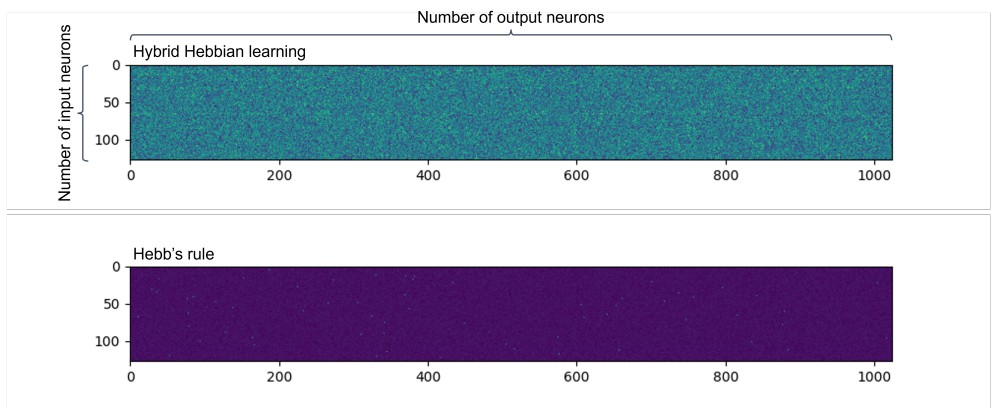

Figure 20: Visualization of weight vectors. (top) the weight vectors in the last layer trained by NeAW Hebbian learning and (bottom) plain Hebb's rule. Dark elements refer the low values while bright elements are with the high values.

## F   Visualization of Weight Vector

Figure 20 shows the weight vectors of the last layer in the encoder trained by different unsupervised learning rules. While plain Hebb's rule in Table 1 achieves the higher variance of the neuron activity than the other learning rules, the variance between context vectors of different objects is low. It indicates that the trained model uses more neurons to represent objects, but it fails to activate the different subset of neurons for different objects. Given plain Hebb's rule repeatedly strengthens the weight of frequently activated pre- and post- synaptic neurons, the weight vectors will mostly learn the commonly appeared position vectors of objects, which degrade the sample-to-sample variation. Figure 20 describes the sparse weight matrix trained by Hebb's rule showing that the only few elements are with high values.

## G   Biological Plausibility of NeAW Learning

As the key feature of NeAW Hebbian learning is to alleviate the biased activity issue, the question is whether our brains have similar functions. We view the NeAW Hebbian learning is analogous to Homeostatic plasticity in a biological brain (Turrigiano & Nelson, 2004). It explains the excitatory and inhibitory neurons are dynamically controlled to relieve the biased activity and help the proper brain function such as memory (Keck et al., 2017). It is still an active research area, but one mechanism in such neurons is to strengthen the inhibitory neurons onto excitatory neurons to reduce the activity, or vice versa. Our NeAW learning rule, which dynamically utilizes Hebbian learning (excitation) and anti-Hebbian learning (inhibition) as a function of the neuron activity, is similar with such neuron dynamics driven by the homeostatic plasticity in the brain. Also, NeAW Hebbian learning is still locally defined unsupervised rule, which is an important feature of biologically plausible learning. However, according to Dale's principle, the mature neuron releases the single type of neurotransmitters indicating the neuron can be either excitatory or inhibitory, while NeAW Hebbian learning assumes the neuron can be both the excitatory and inhibitory but as a function of the activity.

