# OpenReview forum: "Unsupervised 3D Object Learning through Neuron Activity aware Plasticity"
_ICLR.cc/2023/Conference — ICLR 2023 poster_

### Official Review · Reviewer_rCep · 2022-10-25

**Confidence:** 4
**Correctness:** 3
**Technical Novelty And Significance:** 3
**Empirical Novelty And Significance:** 3
**Recommendation:** 8

**Clarity, Quality, Novelty And Reproducibility:**

- The paper is written very clearly with adequate figures as required. The contribution made by the paper to combine biologically plausible unsupervised local learning with deep neural networks -- that typically work well only with gradient-based training -- is novel and presents an interesting direction for future exploration.
- I don't see code uploaded with the submission, I request the authors to please consider uploading the implementation for reproducibility of presented results.


**Strength And Weaknesses:**

Strengths:
1. The authors present a neat combination of a biologically plausible local learning method (Hebbian/Anti-Hebbian networks are backed both by prior computational neuroscience modeling and experimental data on cortico-cortical connections) with deep neural networks for 3D object classification. The proposed NeAW Hebbian learning is clearly more expressive in the low-training data regime as shown in Fig. 7 even in comparison with supervised learning, this is very promising.

2. NeAW quite significantly improves the performance of other Hebbian learning based methods. It is very interesting that this simple yet effective twist on Grossberg's rule results in ~15% and ~20% gains on ModelNet10 and ModelNet40 respectively.

3. This paper is written really well with adequate graphical explanations to distinguish the different Hebbian learning rules from NeAW (Fig. 3 clearly shows how they are different from Hebbian and Anti-hebbian learning). Section 3.3 which is an important part of the paper that introduces NeAW is written particularly well.

Weaknesses:
1. Performance differences between NeAW and supervised methods on the full training set is quite big. Part of this can be explained by the difference in number of parameters between these models (model size). I think it would be fairer to compare NeAW when its model size is increased to match that of PointNet, PointNet++, DGCNN. The paper's contribution would seem even stronger if they match or improve performance of their unsupervised local learning rule with above mentioned supervised counterparts.
2. There seem to be multiple supervised approaches I find at https://paperswithcode.com/sota/3d-point-cloud-classification-on-modelnet40 that achieve >90% on ModelNet40 with as few as 1.1M parameters (~4.4 MB if using fp32). If the authors would like to emphasize the strength of their model with few parameters, they must compare to these methods using very few parameters and yet achieving high performance on ModelNet.
3. This is not necessarily a weakness per se, but I find it odd that the performance is insensitive to choice of a/b. Is it possible that the authors didn't cover a wide enough range to start seeing differences? Given that purely Hebbian or Anti-Hebbian methods work much poorly, setting very low learning rates for either (a or b) might produce significant performance differences. It would be great if the authors could test a wider range of a/b ratio and see if performance differences arise.

**Summary Of The Paper:**

This paper presents an unsupervised method to train deep neural networks using Hebbian/Anti-hebbian learning on 3D object classification from point clouds. The authors highlight (empirically and theoreticalaly) key disadvantages of using just Hebbian or Anti-hebbian learning; they demonstrate the improved expressivity and representational characteristics of using a neural activity based combination of both Hebbian and Anti-hebbian learning. They show that the latter outperforms other Hebbian learning rules on ModelNet10 and ModelNet40 3D object classification tasks.

**Summary Of The Review:**

The paper presents an interesting combination of local learning with Hebbian/Anti-hebbian rules with deep neural networks and demonstrates its utility on 3D object classification on ModelNet10 and ModelNet40. The paper makes novel contributions that overcome key limitations of using Hebbian and Anti-hebbian learning rules in the context of deep learning -- prior computational modeling work has however experimented the combined use of Hebbian and Anti-hebbian learning rules (authors cite relevant work in this direction in Sec 3.3 Theorem 2).

I have two main concerns: 1) Comparison to the current supervised techniques in the paper don't match model size, 2) there are other supervised methods that use very few parameters and achieve high performance on ModelNet40. That said, NeAW makes a significant improvement in using $\textbf{local learning}$ to train deep neural networks. I recommend accept (6/10) at this stage due to the above two evaluation concerns. If the authors could show how their model performs when they increase model size, I may update my score as the current evaluation only uses a small number of parameters with model size 3.1 MB.

---

> ### Author Response · Authors · 2022-11-19
> **Response to the Reviewer rCep 1/3**
>
> We are glad to know that the reviewer considers our method interesting. Below we try to address each of your comments:
>
> > Performance differences between NeAW and supervised methods on the full training set is quite big. Part of this can be explained by the difference in number of parameters between these models (model size). I think it would be fairer to compare NeAW when its model size is increased to match that of PointNet, PointNet++, DGCNN. The paper's contribution would seem even stronger if they match or improve performance of their unsupervised local learning rule with above mentioned supervised counterparts.
>
> **Reply**: We agree with the reviewer that matching the performance of supervised learning by expanding the model will be ideal. However, in general, fully unsupervised rules like ours tend to have lower performance than supervised models when (1) large amount of labelled data is available and (2) large number of classes are involved. Unsupervised models tend to perform better at low labelled data setting as shown in our and other works [1].
>
> Instead of only accuracy, in this paper, we primarily focused on the comparative analysis of unsupervised (our) and supervised learning for *low complexity model architectures common in Tiny-ML*. However, even in this setting, we agree with the reviewer that it is important to study how the model accuracy changes with the model size. Hence, we performed additional experiments to study this trend (see Appendix E).
>
> We design multiple larger and smaller models to understand how the supervised (Backpropagation) and unsupervised (NeAW) model performance change depending on the model complexity. Total 10 different models are compared. Figure 16 in Appendix E is added to show their number of trainable parameters and test accuracy in ModelNet40. The memory size is 0.01MB, 0.02MB, 0.04MB, 0.09MB, 0.17MB, 1.2MB, 1.9MB, 3.1MB (baseline), 5.6MB, and 6.8MB. The five small models are simply designed with a single-layer encoder and classifier to significantly reduce the number of the parameters. The other five models are designed with the same three-layer architecture, but the dimension of the layers differs with the baseline.
>
> For the larger models, we observe that the supervised models show improved test accuracy (82.0%) while the accuracy of the unsupervised models saturate (76.4%). On the other hand, accuracy significantly drops at the small supervised models while the unsupervised models are less sensitive to the model size. The performance difference is particularly large at the model which size is 0.09MB: the unsupervised model achieves 71.7% and the supervised model shows 59.9%.
>
> In summary, we would like to note that the advantage and purpose of NeAW learning is at the good performance in such tiny models, rather than achieving the state-of-the-art accuracy in large scales. However, it would be an important future work to scale up the unsupervised model as many Hebbian-like learning rules have not been applied in the large-scale models.
>
> [1] Lagani, Gabriele, et al. "Hebbian semi-supervised learning in a sample efficiency setting." Neural Networks 143 (2021): 719-731.

---

> > ### Author Response · Authors · 2022-11-19
> > **Response to the Reviewer rCep 2/3**
> >
> > > There seem to be multiple supervised approaches I find at https://paperswithcode.com/sota/3d-point-cloud-classification-on-modelnet40 that achieve >90% on ModelNet40 with as few as 1.1M parameters (~4.4 MB if using fp32). If the authors would like to emphasize the strength of their model with few parameters, they must compare to these methods using very few parameters and yet achieving high performance on ModelNet.
> >
> > **Reply**: We would first like to note that NeAW is an innovation in learning rule, rather than in model architecture. The advantage of NeAW is the unsupervised learning, which enables learning from less labelled data, even for low-complexity models. The alternative model architectures with lower/comparative complexity referred by the reviewer are trained in a supervised fashion.
> >
> > | Model      | ModelNet40 (%) | Memory (MB) | FLOPs / sample|
> > | ----------- | ----------- | ----------- | ----------- |
> > | PointNet      | 89.2       | 40 (13.2 for classification) | 0.44G|
> > | PointNet++   | 91.9        | 12 (5.6 for classification) | 1.68G|
> > | DGCNN|92.9|21 (6.9 for classification)|2.78G|
> > | Ours - NeAW | 76.2 | 3.1 | 0.14G |
> >
> > Based on reviewer’s suggestions, we studied complexity of these existing low-complexity supervised models. An example given by the reviewer is DSPoint (93.5%, 1.16M) at https://paperswithcode.com/sota/3d-point-cloud-classification-on-modelnet40. However, we also observe the authors of DSPoint report the latency is 15.8 times higher in the model than PointNet. Our model has a very similar architecture like PointNet but has much fewer parameters; and hence, has much lower latency than DSPoint. In other words, along with number trainable parameters and accuracy, the computational costs such as FLOPs and latency of the models are also critical quality factors for Tiny-ML models and learning rules.
> >
> > Table above shows a comparative analysis of FLOPS of our models and the supervised models in the paper. We observe that our complexity is about 3 times lower than PointNet and 18.5 times lower than DGCNN showing our model is more suitable for Tiny-ML implementation. In the table, we list the two memory sizes of the models to clarify the entire model size and the classification mode size. We also change the model size in Table 2 from the entire model size to the classification model. We sincerely apologize for our mistake and the confusion.

---

> > > ### Author Response · Authors · 2022-11-19
> > > **Response to the Reviewer rCep 3/3**
> > >
> > > > This is not necessarily a weakness per se, but I find it odd that the performance is insensitive to choice of a/b. Is it possible that the authors didn't cover a wide enough range to start seeing differences? Given that purely Hebbian or Anti-Hebbian methods work much poorly, setting very low learning rates for either (a or b) might produce significant performance differences. It would be great if the authors could test a wider range of a/b ratio and see if performance differences arise.
> > >
> > > **Reply**: We apologize for the confusion between the pure Hebbian (or anti-Hebbian) and a, b settings. Purely Hebbian and anti-Hebbian learning rules in Figures 3 in the paper still indicate NeAW learning but with different values for a and b in Equation (4).   Essentially, “purely Hebbian learning” refers to NeAW learning rule in Equation (4) but with a=1 and b=-1 where the weight update is always positive regardless of the neuron activity. Similarly, the “purely anti-Hebbian learning” indicates a=-1 and b=1 in Equation (4). In other words, “purely Hebbian learning” and “purely anti-Hebbian learning” in Figures indicate always positive or always negative weight update, respectively. To avoid the confusion, we explicitly define “NeAW-H” for purely Hebbian learning and “NeAW-aH” for purely anti-Hebbian learning in the section 3.3. The labels in the related figures are also changed to NeAW-H and NeAW-aH.
> > > In contrast, the very low learning rates for either a or b (Figure 13 in Appendix D)  mean that the weight update is still related to the neuron activity. For example, if a=1 and b=0, we only update the weight of the low-activity neuron while do not update the weight of the high-active neuron. The case where a=0 and b=1 can be similarly understood.
> > >
> > > As the reviewer suggested, we study the performance in a wider range of a and b. We consider the very low rates such as 0, 0.01, 0.05, and 0.1 for the additional candidates of a and b. Figure 13 in Appendix D shows the variance and accuracy with the candidates of a and b. We observe the final accuracy achieved by the very low a and b values are still similar with the others as described in Figure 13(b) and (d). However, we also observe that the variance of the neuron activity slowly increases in the lower rate as shown in Figure 13(a) and (c), indicating the slow relaxation of the biased activity. It can be intuitively understood by the vector schematic in the new figure (Figure 14 in Appendix D). Though the NeAW learning with either a=0 or b=0 will not guarantee the fast relaxation of the biased activity, the direction of the weight update can be still properly designed. Figure 15 in Appendix D is added to show the test accuracy for ModelNet40 with the models which encoder is only trained by 5 epochs. We observe that having higher a and b generally achieves higher accuracy than having very low a and b values.
> > >
> > > > I don't see code uploaded with the submission, I request the authors to please consider uploading the implementation for reproducibility of presented results.
> > >
> > > **Reply**: Thank you for the suggestion. We upload the code implementation in the supplementary material. The code currently does not include the datasets. We will upload the code with pre-trained models, datasets, and manual on github if the paper is accepted.
> > >
> > >
> > > > I have two main concerns: 1) Comparison to the current supervised techniques in the paper don't match model size, 2) there are other supervised methods that use very few parameters and achieve high performance on ModelNet40. That said, NeAW makes a significant improvement in using  to train deep neural networks. I recommend accept (6/10) at this stage due to the above two evaluation concerns. If the authors could show how their model performs when they increase model size, I may update my score as the current evaluation only uses a small number of parameters with model size 3.1 MB.
> > >
> > > **Reply**: We appreciate the reviewer’s feedback which helped us to improve the quality of the paper. Our additional experiments show that NeAW is suitable for unsupervised learning of small models and the performance of a given (small) network trained with NeAW is less sensitive to model size reduction compared to the case when the same network is trained with supervised learning. We also observe that, compared to other supervised networks of small model size, our model has less computational complexity (less FLOPS and latency). On the other hand, our current experiments also show that NeAW-based learning may not improve performance of large models.

---

> > > > ### Author Response · Authors · 2022-11-24
> > > > **Follow-up discussion with the Reviewer rCep**
> > > >
> > > > Dear Reviewer rCep,
> > > >
> > > > Thank you again for your time and effort to improve the quality of the paper. We hope your concerns have been adequately addressed in the rebuttal. We would really appreciate it if you could please give us some feedback on the rebuttal. We will be more than happy to further discuss this with you if you have any other concerns regarding our current draft and the rebuttal.
> > > >
> > > > Best,
> > > >
> > > > Paper3864 authors.

---

> > > > ### Comment · Reviewer_rCep · 2022-11-26
> > > > **Satisfied by author response addressing comments in my review**
> > > >
> > > > Dear authors,
> > > >
> > > > I apologize for the slight delay in my update and I thank you very much for the detailed response to the comments I raised in my review.
> > > >
> > > > `..On the other hand, accuracy significantly drops at the small supervised models while the unsupervised models are less sensitive to the model size. The performance difference is particularly large at the model which size is 0.09MB: the unsupervised model achieves 71.7% and the supervised model shows 59.9%.`
> > > >
> > > > This is awesome! I'm really impressed that the unsupervised model is quite robust to low model sizes. Thank you also
> > > >
> > > > `We would first like to note that NeAW is an innovation in learning rule, rather than in model architecture. The advantage of NeAW is the unsupervised learning, which enables learning from less labelled data, even for low-complexity models. The alternative model architectures with lower/comparative complexity referred by the reviewer are trained in a supervised fashion.`
> > > >
> > > > I agree, I believe NeAW is trying to push for advances in both TinyML and unsupervised representation learning. Hence, I was recommending a holistic evaluation that also includes supervised models. However, I acknowledge that the proposed NeAW learning rule is a meaningful improvement to existing unsupervised Hebbian/AntiHebbian learning rules.
> > > >
> > > > I also appreciate the additional data and figures to study the effect of a and b. I would like to make a small suggestion to improve the clarity of this plot. I would suggest using a suitable colormap that highlights the low and high values of a and b in Fig. 13 (a), (c).
> > > >
> > > > Thank you also for committing to upload the code on GitHub if the paper is accepted.
> > > >
> > > > As my response to the author rebuttal above suggests, I am more convinced than before of the value of the contribution of the NeAW learning rule to TinyML and unsupervised (Hebbian) representation learning. I am hence raising my score to 8 and recommend acceptance of this paper. I think this paper presents some really key contributions to produce low-cost unsupervised machine learning models.

---

> > > > > ### Author Response · Authors · 2022-11-26
> > > > > **Thank you for the positive feedback**
> > > > >
> > > > > Dear Reviewer rCep,
> > > > >
> > > > > We are very happy to know that you are satisfied with our response. We agree with your additional suggesion, and will improve Figure 13(a) and (c) to clarify the range of a and b using more colors.
> > > > >
> > > > > We appreciate the updated rating. Also, we realize "Flag For Ethics Review" is updated. We would like to kindly ask whether we make unprofessional behaviors during the rebuttal. Thank you so much again for your positive feedback and effort to improve the quality of our paper.
> > > > >
> > > > > Best,
> > > > >
> > > > > Paper3864 authors.

---

> > > > > > ### Comment · Reviewer_rCep · 2022-11-26
> > > > > > **Follow up on update after author response**
> > > > > >
> > > > > > I apologize for the inconvenience, it looks like one of the checkboxes under ethics review got selected accidentally while I was updating the score. I have corrected this error on my end.
> > > > > >
> > > > > > Thank you for pointing this out.

---

### Official Review · Reviewer_oMed · 2022-10-31

**Confidence:** 3
**Correctness:** 4
**Technical Novelty And Significance:** 3
**Empirical Novelty And Significance:** 2
**Recommendation:** 6

**Clarity, Quality, Novelty And Reproducibility:**

**Quality**
The theoretical contributions in the paper are solid. Empirically, there are a few concerns as highlighted above. One of the main concerns with the paper is how the contributions are framed; if the paper's contributions are appropriately framed, its significance will become more clear.

**Clarity**
The paper is generally well-written and the figures are particularly well illustrated. Some minor comments:

- A parenthesis appears to be missing from Eqn 2
- The paper has a few typos

**Originality**
It is unclear to what extent NeAW is different from other Hebbian-like learning rules proposed in the past. Is this the first Hebbian-like rule to avoid the activity concentration problem illustrated in Figure 2? If so this would be a quite significant contribution, but it would also require the authors to perform experiments with more learning rules to demonstrate this.

**Strength And Weaknesses:**

**Strengths**
The learning rule proposed by the authors is interesting, and some theoretical properties about it are proven. In my view, the most compelling point of this paper is the design of a learning rule that can produce strong representations of the data in an unsupervised way (as illustrated by Figures 4 and 5). Moreover, the theoretical analysis appears solid. A number of experiments are presented showing the properties of NeAW. The paper is also generally well written and easy to understand.

**Weaknesses**
The motivation for the paper is unfortunately a little unclear to me. What is the purpose of the NeAW rule applied to the 3D point cloud task? Do the authors wish to show good performance on the 3D point cloud task under model size constraints (as suggested by Table 2)? Do the authors wish to show good performance in low-data settings (as suggested by Figure 7)? Alternatively, is the main motivation to demonstrate an unsupervised learning rule that can balance neuron activity (i.e. Figure 2), with the 3D point cloud task as a case study?

If the motivation is to solve the 3D point cloud task more effectively, I would suggest emphasizing the relative advantages of NeAW more clearly throughout the paper. If the motivation is to propose a better Hebbian learning rule, I would suggest focusing the discussion and experiments on other similar Hebbian rules to emphasize that NeAW is superior. I would also suggest commenting on the biological significance of the rule since Hebbian learning rules have traditionally been proposed as models of biological neuron learning (e.g. to what extent can NeAW be considered biologically plausible?).

In terms of the specific motivation of NeAW rule, section 3.2 suggests that the poor distribution of neuron activity in Hebbian and anti-Hebbian learning rules leads to poor performance, and that NeAW achieves better performance by creating a more uniform activity distribution. While it is clear that NeAW achieves a more uniform activity distribution and higher performance, the causal link between these two is not clear to me unfortunately. Can the authors demonstrate empirically that a more uniform activity distribution *causes* higher performance (right now only a correlation is demonstrated)?

I also have some concerns regarding the experiments. From Table 2, the paper argues that NeAW achieves good performance despite a lower model size. However, the model size seems linked to the model architecture used in the paper and not NeAW. If the authors wish to show that NeAW can be used to train small sized models more effectively than comparable learning rules, they may want to apply all the learning rules in Table 2 to a single (small) model architecture and evaluate performance.


**Summary Of The Paper:**

The paper proposes an unsupervised learning rule, NeAW, combining Hebbian and anti-Hebbian learning and applies it to 3D point cloud classification. The learning rule is motivated by the observation that Hebbian and anti-Hebbian learning don't produce diverse representations across neurons (there is low variance in neuron activity). The authors find that NeAW outperforms Hebbian and anti-Hebbian learning, achieves high performance with fewer parameters than comparable models, and outperforms backprop in low-data settings.

**Summary Of The Review:**

The paper proposes an interesting Hebbian learning rule with some solid theoretical analysis. However, there are some concerns with the experiments and especially with the motivation of the paper as noted above. It would also be great if the authors could further clarify the novelty of the proposed rule.

---

> ### Author Response · Authors · 2022-11-19
> **Response to the Reviewer oMed 1/4**
>
> We are grateful to the reviewer for their constructive feedback about our work. We hope we have addressed their concerns and questions regarding the paper and hope they will reconsider their rating.
>
> > The motivation for the paper is unfortunately a little unclear to me. What is the purpose of the NeAW rule applied to the 3D point cloud task? Do the authors wish to show good performance on the 3D point cloud task under model size constraints (as suggested by Table 2)? Do the authors wish to show good performance in low-data settings (as suggested by Figure 7)? Alternatively, is the main motivation to demonstrate an unsupervised learning rule that can balance neuron activity (i.e. Figure 2), with the 3D point cloud task as a case study?
>
> > If the motivation is to solve the 3D point cloud task more effectively, I would suggest emphasizing the relative advantages of NeAW more clearly throughout the paper. If the motivation is to propose a better Hebbian learning rule, I would suggest focusing the discussion and experiments on other similar Hebbian rules to emphasize that NeAW is superior.
>
> **Reply**: The key motivation of this paper is to develop theoretical foundations of an unsupervised learning rule that can balance the neuron activity to improve (unsupervised) clustering quality and hence, classification accuracy. The key contribution is to empirically and theoretically show that the NeAW learning allows more neurons to selectively activate depending on the target classes, thereby, effectively clustering various classes in the encoded space. The empirical results are presented using object classification from 3D point cloud as an example task to demonstrate three outcomes of improved learning (and clustering), namely, (1) the higher accuracy in the classification task than the prior Hebbian-like learning rules, (2) enabling good accuracy with small model architectures, and (3) improve accuracy in low-data settings than end-to-end supervised learning.
>
> In addition to the three outcomes, we improve the discussion in the section 4 and experiments such as t-SNE plots (Figure 4) and the neuron activity distribution (Appendix C) for the other Hebbian learning rules to effectively show that NeAW is superior to them. Also, we discuss the casual relation between the biased activity and poor representation learning in detail throughout the section 4 and Appendix B. In section 3.2, we again clarify the main motivation.
>
> > I would also suggest commenting on the biological significance of the rule since Hebbian learning rules have traditionally been proposed as models of biological neuron learning (e.g. to what extent can NeAW be considered biologically plausible?).
>
> **Reply**: Thanks for the interesting question. We would like to stress that our objective is to develop an improved unsupervised learning rule for machine learning tasks; we do not aim to implement a biologically plausible learning in the paper. However, we agree with the reviewer that, it is interesting to understand whether there exists any biological relevance and plausibility. We observe that the NeAW Hebbian learning is similar to Homeostatic plasticity in a biological brain [1]. It indicates the excitatory and inhibitory neurons are dynamically controlled to relieve the biased activity and help the proper brain function such as memory [2]. It is still an active research area, but one mechanism in such neurons is to strengthen the inhibitory neurons onto excitatory neurons to reduce the activity, or vice versa. Our NeAW learning rule, which dynamically utilizes Hebbian learning (excitation) and anti-Hebbian learning (inhibition) as a function of the neuron activity, is similar with such neuron dynamics driven by the homeostatic plasticity in the brain. Also, NeAW Hebbian learning is still locally defined unsupervised rule, which is an important feature of biologically plausible learning. We add a comment on the biological plausibility in section 3 and add Appendix G to discuss the biological plausibility of the NeAW Hebbian learning.
>
> [1] Turrigiano, Gina G., and Sacha B. Nelson. "Homeostatic plasticity in the developing nervous system." Nature reviews neuroscience 5.2 (2004): 97-107.
>
> [2] Keck, Tara, et al. "Integrating Hebbian and homeostatic plasticity: the current state of the field and future research directions." Philosophical Transactions of the Royal Society B: Biological Sciences 372.1715 (2017): 20160158.

---

> > ### Author Response · Authors · 2022-11-19
> > **Response to the Reviewer oMed 2/4**
> >
> > > In terms of the specific motivation of NeAW rule, section 3.2 suggests that the poor distribution of neuron activity in Hebbian and anti-Hebbian learning rules leads to poor performance, and that NeAW achieves better performance by creating a more uniform activity distribution. While it is clear that NeAW achieves a more uniform activity distribution and higher performance, the causal link between these two is not clear to me unfortunately. Can the authors demonstrate empirically that a more uniform activity distribution causes higher performance (right now only a correlation is demonstrated)?
> >
> > **Reply**: We thank for this constructive comment. We did not intend to claim a causal relation between balanced activity and better performance; we sincerely apologize that the text was not clear on this aspect. To avoid any confusion, we have now added a detailed analysis of the causal and correlated relations between the activity (biased and balanced) and the performance (Appendix B). Our conclusion is that the biased activity *causes* the poor performance, but balancing the activity is *correlated* with a higher performance. Specifically, we show that NeAW improves learning improves because it leads to *balanced activity with object-dependent activation*; only balancing the activity (without object-dependent activation) may not improve performance. We have also edited section 3.2 and rest of the paper to clearly state this observation. The following discussions elaborates our approach to establish the previous observations.
> >
> > **1) causal relation between the biased activity and poor learning**: Hebbian-like unsupervised learning rules with Winner-Take-All (WTA) can be interpreted as a clustering module. We focus on the factors that affect the quality of clustering to build a causal connection between the neuron activity and the performance. It is well-known that (1) the number of cluster centroids and (2) their separability play a significant role in determining the quality of clustering [1]. If there are too few cluster centroids, irrelevant points will be associated with the same cluster centroid. Likewise, if the cluster centroids are not separable, i.e., multiple centroids are positioned at the same location, the effective number of clusters will be still low. In the end, we need the *enough number of well separable cluster centroids that behave as principal components* for different local features in the object. In Corollary 2.1 (Appendix A), we have proved that Hebbian and anti-Hebbian learning leads to biased neuron activity and hence, fewer principal components than NeAW learning. Fewer principal components indicate fewer separable cluster centroids which leads to poor performance. Hence, using Corollary 2.1 we can state that a biased activity in the Hebbian and anti-Hebbian learning causes a poor performance.
> >
> > That is, the biased activity problem should be resolved to improve the performance. However, it does not necessarily indicate that the balanced activity leads to a better performance. We apologize for the confusion and modify the texts that may imply the causality between the balanced activity and a high performance.

---

> > > ### Author Response · Authors · 2022-11-19
> > > **Response to the Reviewer oMed 3/4**
> > >
> > > **2) correlation between the balanced activity and better clustering**: The causal link between biased activity and poor performance suggests that reducing activity bias may improve performance. However, it does not necessarily indicate that the only balancing activity causes a better performance. In particular, we observe that the “balanced activity with object-dependent activation” is correlated with the clustering quality. The idea behind is that the neurons that activate regardless of the object classes (no object dependency) will not capture the features to design the classification boundary. In this light, we design an additional experiment to quantify the change of the clustering quality depending on how dependent neurons are to different object classes. We quantify the quality using the dissimilarity matrix. The dissimilarity matrix for ModelNet10 is given by 10x10 matrix, where each element is the dissimilarity between the encoded vectors of the corresponding two object classes (Figure 7(a) in Appendix B). We choose cosine similarity to calculate the dissimilarity. Let xA and xB as the encoded vector for A and B objects. Then, the dissimilarity in the matrix (D) is given by D[A][B] = 1 – (xA dot xB) / (|xA||xB|). We calculate the change of the L2 norm of the dissimilarity matrix by deactivating neurons one by one. The experiment results demonstrate that deactivating the neuron that object-dependently activates reduces the dissimilarity between the objects, indicating a poor clustering quality. In contrast, deactivating the neuron that activates regardless of the object labels increases the dissimilarity, leading to a better clustering quality (Figure 7(b) in Appendix B). Note, the number of activating neurons during the experiment are same as we deactivate a single neuron at a time.
> > >
> > > In summary, the biased activity causes a poor learning in the proposed model while the balanced activity is not a cause of the better performance. The additional experiment demonstrates that the clustering quality varies depending on both the activity balance and the neuron’s object dependency.
> > >
> > > [1] Kodinariya, Trupti M., and Prashant R. Makwana. "Review on determining number of Cluster in K-Means Clustering." International Journal 1.6 (2013): 90-95.
> > >
> > > > I also have some concerns regarding the experiments. From Table 2, the paper argues that NeAW achieves good performance despite a lower model size. However, the model size seems linked to the model architecture used in the paper and not NeAW. If the authors wish to show that NeAW can be used to train small sized models more effectively than comparable learning rules, they may want to apply all the learning rules in Table 2 to a single (small) model architecture and evaluate performance.
> > >
> > > **Reply**: Table 2 discusses three learning rules: unsupervised, supervised, and self-supervised. We have already studied supervised (backpropagation) and unsupervised (NeAW) learning for small models (Table 2). Based on the reviewer’s suggestion, to complete the study we augment our current model to apply self-supervised learning model (Figure 17 in Appendix E). A simple decoder with three deconvolution and upsample layers is added in the proposed architecture to create an autoencoder architecture. The self-supervised training is applied between encoder and reconstruction decoder to learn the latent representation using a Chamfer distance loss between ground truth and reconstructed point cloud. The parameters of the encoder are fixed after the self-supervised learning and the classifier decoder is trained via supervised learning (as in our NeAW model). We observe the self-supervised model achieves the similar accuracy (92.2% for ModelNet10 and 79.5% for ModelNet40) as NeAW based unsupervised model (88.2% for ModelNet10 and 76.2% for ModelNet40). While the encoder and classifier used in the inference are same with the supervised and unsupervised models (3.1MB), the number of parameters even in the simple reconstruction decoder is appreciable (263,539, 1.0MB) resulting in an overall larger model size (total 4.1MB). The activation size per sample is also increased significantly (from 808 to 12,104) due to the need for reconstruction during training. We add the self-supervised model based on our proposed architecture in Table 2.
> > >
> > > We also explore how the accuracy of the supervised and NeAW learning decreases with even smaller model size by designing simpler models with single-layer encoder and classifier. We observe the accuracy for the ModelNet40 task of the simpler supervised models drops (46.6% - 61.9%) more than the unsupervised models (56.1% - 72.5%) as shown in the new figure (Figure 16 in Appendix E).

---

> > > > ### Author Response · Authors · 2022-11-19
> > > > **Response to the Reviewer oMed 4/4**
> > > >
> > > > > Some minor comments: A parenthesis appears to be missing from Eqn 2. The paper has a few typos.
> > > >
> > > > **Reply**: we apologize for the mistakes in the paper. The parenthesis and typos are addressed
> > > >
> > > > > It is unclear to what extent NeAW is different from other Hebbian-like learning rules proposed in the past. Is this the first Hebbian-like rule to avoid the activity concentration problem illustrated in Figure 2? If so this would be a quite significant contribution, but it would also require the authors to perform experiments with more learning rules to demonstrate this.
> > > >
> > > > **Reply**: The key contribution of the paper is to present a theoretical analysis and empirical results to demonstrate that the neuron activity aware learning leads to the balanced neuron activity and improved clustering in the latent space, thereby improving performance for classification task even for simpler model.
> > > >
> > > > Along with the NeAW learning with only positive feedback (Hebbian) and only negative feedback (anti-Hebbian), we have also compared the results with other Hebbian learning rules such as plain Hebb’s rule, Grossberg’s rule, and Oja’s rule. As mentioned in the paper, these Hebbian learning rules does not effectively reduce the biased neuron activity, and so, does not create more separable clusters. This can be shown using the t-SNE plots for the latent space, and now we have added t-SNE plots for Hebb’s rule, Grossberg’s and Oja’s rule (Figure 4). We observe the several differences between NeAW and these learning rules. First, these rules do not incorporate the activity or the function of the activity. In Equation (1) in the paper, they only depend on the input and output values without the information of how frequent the output neuron activates. The variants of the plain Hebb’s rule aim to stabilize the unstable positive feedback issue, but it does not necessarily mean the activity is balanced as shown in Table 1. Next, the NeAW learning rule is dynamically controlled by the activity, but these learning rules are constant as the parameters of the rules are fixed. Note, biological synaptic plasticity is a dynamical characteristic as a complex function of neuromodulators, rather than a constant characteristic. Recently, Hebbian ABCD rule has attracted many attentions and widely used in reinforcement learning tasks. While the application is different with ours, it offers more freedom to design the weight update rule by using the functions of input alone, output alone, and the multiplication of input and output together. However, the parameters in the rule (A, B, C, and D) are trained by end-to-end supervised learning, which is orthogonal to our fully unsupervised approach. There is a related prior work by B. Kang et al. in the reference to avoid the biased activity problem. However, this work does not show the theoretical analysis at all and is evaluated on 2D datasets.
> > > >
> > > > We perform more experiments to compare the NeAW learning with the prior Hebbian learning rules, Hebb’s rule, Grossberg’s rule, and Oja’s rule. The neuron activity distribution is visualized in the new figures (Figure 9 and 10 in Appendix C) to show whether the activity is sparse in these learning rules as well. We observe that the activity in these learning rules is still biased than the NeAW learning. While the Hebb’s rule shows relatively balanced activity than the others in Figure 9, many neurons activate regardless of the object classes as shown in Figure 10. Thus, we further demonstrate that the prior Hebbian learning rules, Hebb’s rule, Grossberg’s rule, and Oja’s rule, do not effectively avoid the activity concentration problem and create more separable clusters in the latent space.
> > > >
> > > > > The paper proposes an interesting Hebbian learning rule with some solid theoretical analysis. However, there are some concerns with the experiments and especially with the motivation of the paper as noted above. It would also be great if the authors could further clarify the novelty of the proposed rule.
> > > >
> > > > **Reply**: We are glad to know that the reviewer considers our method interesting. We hope that our response and the revised paper addressed the reviewer’s concerns. We thank the reviewer again for their time and effort to improve the quality of our paper.

---

> > > > > ### Comment · Reviewer_oMed · 2022-11-21
> > > > > **Thank you for your response!**
> > > > >
> > > > > I appreciate the authors' detailed response and additional experiments.
> > > > >
> > > > > Thank you for clarifying the motivation of the paper. Since the primary goal of the paper is to propose a local unsupervised learning rule that balances neuron activity to achieve better performance, I think demonstrating the link between activity balance and performance is key. The experiments in Appendix B and C are quite valuable. However, it is still not clear to me why NeAW outperforms Hebb's rule given the relatively uniform activation distribution for Hebb's rule (Figure 9a). The authors also state that balanced activity merely correlates with better clustering and performance, not necessarily causing it. Thus, the better performance of NeAW is still unexplained. Establishing *why* NeAW outperforms its baselines is critical in my view. I am prepared to raise my rating if this concern can be adequately addressed.
> > > > >
> > > > > Since the authors wish to show that NeAW outperforms other Hebbian rules in general, I also think showing experiments outside the 3D point cloud setting will be important. Otherwise, I think the significance of the empirical results may be limited to this specific setting (although the theory (Theorems 1 and 2) is generally applicable).
> > > > >
> > > > > I also am glad to see the additional experiments in Appendix E and the discussion of biological plausibility in Appendix G.

---

> > > > > > ### Author Response · Authors · 2022-11-23
> > > > > > **Response to follow-up questions 1/3**
> > > > > >
> > > > > > > However, it is still not clear to me why NeAW outperforms Hebb's rule given the relatively uniform activation distribution for Hebb's rule (Figure 9a). The authors also state that balanced activity merely correlates with better clustering and performance, not necessarily causing it. Thus, the better performance of NeAW is still unexplained. Establishing why NeAW outperforms its baselines is critical in my view.
> > > > > >
> > > > > > **Reply**: Thank you for the important feedback. The key to the better performance of NeAW are *the object-dependent activation along with the balanced activity*. As we discussed in the revised submission, compared to other Hebbian-like rule, the NeAW improves clustering quality by creating *a higher number of well-separable cluster centroids that behave as principal components* for different local features in the object. The imbalance in activity (i.e., fewer neuron activation) leads to fewer cluster centroids; NeAW reduces this bias (balanced activity) leading to more cluster centroids. However, only more centroids are not sufficient; we also need higher separation between these centroids to ensure same neurons are not activated regardless of the object class. NeAW improves separability of these cluster centroids by ensuring object-dependent activation of the neurons.
> > > > > >
> > > > > > We apologize if the importance of both factors, namely, object-dependent activation and balanced activity, in resulting in higher performance of NeAW compared to other Hebbian rules are not clearly stated. We have performed additional analysis to clarify this (as shown below) and will update the text with these additional discussions if the paper is accepted.
> > > > > >
> > > > > > **Object-dependent activation in other Hebbian learning**: As the reviewer pointed out, a good example of why balanced activity is not sufficient is Hebb’s rule where the neuron activity is evenly distributed (Figure 9(a)) but classification performance is poor. In addition to Figure 9(a) describing the activity balance, it is important to see Figure 12(a) that shows the average activation for *each object class* in Hebb’s rule. The column in the figure represents in which object classes the neuron frequently activates (red elements). Compare to that of NeAW learning in Figure 10, we observe that the most columns are dense with red in Figure 12(a), indicating many neurons activate regardless of the object in Hebb’s rule. **As we view the neuron activation in WTA modules as the cluster centroid, it means Hebb’s rule creates many cluster centroids that group all the objects regardless of their labels, thereby, decreasing the dissimilarity between the objects in the latent space.** In other words, it is hard to distinguish the rows (i.e. encoded vectors) in Figure 12(a). This is the reason why Hebb’s rule gives a poor performance while having the balanced activity. Though Grossberg’s rule and Oja’s rule in Figure 12(b) and (c) display sparse columns and more object-dependent activation, the number of activating neurons is small (~35). In this case, they create separable cluster centroids but the number of them are too few. As discussed in the rebuttal, we need *the enough number of well separable cluster centroids that behave as principal components* for different local features in the object. However, the qualitative results display that the baseline learning rules only create few separable cluster centroids. It is parallel with Corollary 2.1, which explains that applying only Hebbian learning leads to a poor selection of the principal components than NeAW.

---

> > > > > > > ### Author Response · Authors · 2022-11-23
> > > > > > > **Response to follow-up questions 2/3**
> > > > > > >
> > > > > > > *Table. Dissimilarity matrix in NeAW learning (below)*
> > > > > > > |             | Monitor   | Totilet   | Dresser   | Table   | Night stand   | Bed    | Bathtub   | Desk   | Chair   | Sofa   |
> > > > > > > |:-------------:|:-----------:|:-----------:|:-----------:|:---------:|:---------------:|:--------:|:-----------:|:--------:|:---------:|:--------:|
> > > > > > > | Monitor     | 0.00      | 0.13      | 0.18      | 0.25    | 0.20          | 0.25   | 0.28      | 0.15   | 0.26    | 0.20   |
> > > > > > > | Toilet      | 0.13      | 0.00      | 0.07      | 0.17    | 0.09          | 0.13   | 0.15      | 0.09   | 0.08    | 0.11   |
> > > > > > > | Dresser     | 0.18      | 0.07      | 0.00      | 0.18    | 0.02          | 0.18   | 0.14      | 0.09   | 0.14    | 0.14   |
> > > > > > > | Table       | 0.25      | 0.17      | 0.18     | 0.00    | 0.19          | 0.11   | 0.15      | 0.06   | 0.27    | 0.11   |
> > > > > > > | Night stand | 0.20      | 0.09      | 0.02      | 0.19    | 0.00          | 0.18   | 0.17      | 0.09   | 0.13    | 0.17   |
> > > > > > > | Bed         | 0.25      | 0.13      | 0.18      | 0.11    | 0.18          | 0.00   | 0.05      | 0.09   | 0.16    | 0.04   |
> > > > > > > | Bathtub     | 0.28      | 0.15      | 0.14      | 0.15    | 0.17          | 0.05   | 0.00      | 0.11   | 0.19    | 0.07   |
> > > > > > > | Desk        | 0.15     | 0.09      | 0.09      | 0.06    | 0.09          | 0.09   | 0.11      | 0.00   | 0.18    | 0.06   |
> > > > > > > | Chair       | 0.26      | 0.08      | 0.14      | 0.27    | 0.13          | 0.16   | 0.19      | 0.18   | 0.00    | 0.16   |
> > > > > > > | Sofa        | 0.20      | 0.11      | 0.14      | 0.11    | 0.17          | 0.04   | 0.07      | 0.06   | 0.16    | 0.00   |
> > > > > > >
> > > > > > > *Table. Dissimilarity matrix in Hebb's rule (below)*
> > > > > > > |             | Monitor   | Totilet   | Dresser   | Table   | Night stand   | Bed    | Bathtub   | Desk   | Chair   | Sofa   |
> > > > > > > |:-------------:|:-----------:|:-----------:|:-----------:|:---------:|:---------------:|:--------:|:-----------:|:--------:|:---------:|:--------:|
> > > > > > > | Monitor     | 0.00      | 0.06      | 0.06      | 0.14    | 0.06          | 0.11   | 0.13      | 0.07   | 0.09    | 0.10   |
> > > > > > > | Toilet      | 0.06      | 0.00      | 0.01      | 0.08    | 0.01          | 0.04   | 0.05      | 0.03   | 0.03    | 0.04   |
> > > > > > > | Dresser     | 0.06      | 0.01      | 0.00      | 0.07    | 0.01          | 0.03   | 0.05      | 0.02   | 0.05    | 0.03   |
> > > > > > > | Table       | 0.14      | 0.08      | 0.07      | 0.00    | 0.08          | 0.04   | 0.08      | 0.03   | 0.15    | 0.07   |
> > > > > > > | Night stand | 0.06      | 0.01      | 0.01      | 0.08    | 0.00          | 0.05   | 0.06      | 0.03   | 0.03    | 0.05   |
> > > > > > > | Bed         | 0.11      | 0.04      | 0.03      | 0.04    | 0.05          | 0.00   | 0.02      | 0.02   | 0.09    | 0.02   |
> > > > > > > | Bathtub     | 0.13      | 0.05      | 0.05      | 0.08    | 0.06          | 0.02   | 0.00      | 0.06   | 0.09    | 0.01   |
> > > > > > > | Desk        | 0.07      | 0.03      | 0.02      | 0.03    | 0.03          | 0.02   | 0.06      | 0.00   | 0.08    | 0.04   |
> > > > > > > | Chair       | 0.09      | 0.03      | 0.05      | 0.15    | 0.03          | 0.09   | 0.09      | 0.08   | 0.00    | 0.09   |
> > > > > > > | Sofa        | 0.10      | 0.04      | 0.03      | 0.07    | 0.05          | 0.02   | 0.01      | 0.04   | 0.09    | 0.00   |
> > > > > > >
> > > > > > > **Impact of object-dependent activation on dissimilarity**: We quantify the clustering quality using the dissimilarity matrix in Hebb’s rule, Grossberg’s rule, and Oja’s rule to further understand the impact of the object-dependent activation. The dissimilarity is calculated by the same way in the rebuttal and Appendix B. Table above shows that the dissimilarity matrix in NeAW and Hebb’ rule, and all the table elements in NeAW are higher than Hebb’s rule. It indicates that the object classes are easily distinguishable in NeAW. **As many neurons activate regardless of the objects in Hebb’s rule, the inner product between the encoded vectors increases, and the cosine-based dissimilarity decreases leading to the poor clustering quality.** That is, the encoded vectors with less common activation will have the higher dissimilarity, which indicates the better clustering quality. In the end, the higher dissimilarity is preferred; and we consider the L2 norm of the dissimilarity matrix to compare the cluster quality between the learning rules. The L2 norm for Hebb’s rule, Grossberg’s rule, Oja’s rule, and NeAW learning are 0.630, 1.049, 1.313, and 1.482, respectively. Note, the accuracy of them are 72.22%, 74.45%, 80.40%, and 88.22%, respectively.
> > > > > > >
> > > > > > > We think the reviewer’s feedback is important to improve the quality of the paper; and thus, the discussion and experiments here will be added into the manuscript after the revision.

---

> > > > > > > > ### Author Response · Authors · 2022-11-23
> > > > > > > > **Response to follow-up questions 3/3**
> > > > > > > >
> > > > > > > > > Since the authors wish to show that NeAW outperforms other Hebbian rules in general, I also think showing experiments outside the 3D point cloud setting will be important.
> > > > > > > >
> > > > > > > > **Reply**: Thanks for the comment. As the current results are based on 3D object learning, we agree that more empirical results on other task will be important. We choose 2D object learning as an example to validate whether NeAW can learn the object features in both 2D and 3D settings.
> > > > > > > >
> > > > > > > > *Table. test accuracy of 2D point MNIST dataset in various learning methods (below)*
> > > > > > > > |Method    |accruacy (%)   |
> > > > > > > > |:-------------:|:-----------:|
> > > > > > > > | Backpropagation | 96.31 |
> > > > > > > > | NeAW | 94.80 |
> > > > > > > > | Hebb's rule | 58.70 |
> > > > > > > > | Grossberg's rule | 55.21 |
> > > > > > > > | Oja's rule | 69.04 |
> > > > > > > >
> > > > > > > >
> > > > > > > > We design a simple 2D object classification task on point MNIST in Appendix E, but the current paper only includes the result by Backpropagation and NeAW. We perform the additional experiment with Hebb’s rule, Grossberg’s rule, and Oja’s rule. Table above shows the test accuracy of point MNIST datasets in other Hebbian learning rules. The empirical result demonstrate NeAW still outperforms the others in 2D learning task, indicating NeAW is superior to them in learning the structure of both the 2D and 3D objects. We will add the discussion and results in Appendix after the revision. We agree with the reviewer’s view, and further validation with more various model architectures and datasets would continue to study additional datasets.
> > > > > > > >
> > > > > > > > Thank you again for your time and effort to improve tour paper. We look forward to discussing the additional feedback if the answers do not fully address the reviewer’s concern.

---

> > > > > > > > > ### Author Response · Authors · 2022-11-25
> > > > > > > > > **Follow-up discussion with the Reviewer oMed**
> > > > > > > > >
> > > > > > > > > Dear Reviewer oMed,
> > > > > > > > >
> > > > > > > > > We would really appreciate it if you could please give us some feedback on whether or not our response has addressed your concerns. We will be more than happy to further discuss this with you if you have any other concerns regarding our current draft and the rebuttal. We want to thank you again for your valuable time and feedback to improve the quality of the paper.
> > > > > > > > >
> > > > > > > > > Best,
> > > > > > > > >
> > > > > > > > > Paper3864 authors.

---

> > > > > > > > > > ### Comment · Reviewer_oMed · 2022-11-30
> > > > > > > > > > **Thank you for your response!**
> > > > > > > > > >
> > > > > > > > > > I appreciate the authors' detailed response and additional experiments. At this point, most of my concerns have been addressed. I am now inclined to accept this paper and have updated my recommendation accordingly.

---

> > > > > > > > > > > ### Author Response · Authors · 2022-11-30
> > > > > > > > > > > **Thank you for the positive feedback!**
> > > > > > > > > > >
> > > > > > > > > > > We really appreciate your feedback! We will incorporate the follow-up discussion into the final version. Thank you again for your time and effort to improve the quality of the paper.

---

### Official Review · Reviewer_Sia3 · 2022-11-03

**Confidence:** 3
**Correctness:** 3
**Technical Novelty And Significance:** 2
**Empirical Novelty And Significance:** 2
**Recommendation:** 8

**Clarity, Quality, Novelty And Reproducibility:**

In general, the paper is well written. The motivations are clearly explained. The experimental design is also sound. One novelty of this paper is that it added a reference output value to decide if Hebbian and anti-Hebbian learning rule should be applied such that the resulting activities are more evenly distributed across output neurons, which facilitate better learning. The source code is not provided; thus the implementation is not considered for this review.

Some minor typos and unclear phrasing.
- Page 2, Equation (1). The $\Delta$ before w(t+1) should be removed.
- Page 4, Definition 1. The phrasing is not very clear.


**Strength And Weaknesses:**

**Strength**:

In general, this paper proposed an elegant extension of the Grossberg’s rule to address the biased activity problem for both Hebbian and anti-Hebbian learning rules. Some of the strengths of this work are listed below:

- To facilitate the study of different Hebbian learning rules for 3D object classification, the author proposed a model with an encoder containing WTA modules at each layer.

- The author identified the problem of biased activity (a few neurons are activated most of the time regardless of the object) with both Hebbian and anti-Hebbian learning in 3D object classification with their proposed model and linked this observation to the loss of local features in complex 3D objection recognition tasks.

- To remove this bias, NeAW uses Hebbian rule when the output neuron activity is smaller than the average neuron activity p*=1/#output-neurons and uses anti-Hebbian when the activity is larger than p*. Intuitively, this seems like an implicit regularizor to reduce the activities of more activated neurons and increase the activity of less activated ones, and the activities are more evenly distributed across output neurons of a specific layer.

- To illustrate this idea, the authors provided intuitive illustration to understand the geometry of the problem that NeAW is trying to address (Figure 3). And they also claimed to have proved NeAW relieves the biased activity under a given geometric condition.

- The authors also empirically showed that the proposed NeAW results in a better feature representation which leads to better performance compared to other Hebbian-based rules and can even outperform end-to-end supervised learning when data is limited.

**Weaknesses**

- In the motivation study, the authors identified the biased activity problem with Hebbian-based learning rules and argued that a relaxation of this skewed activity is the key to improve learning. In addition to observing the activities resulting from Hebbian learning, I would also suggest studying if end-to-end learning with better classification performance would result in similar biased activities.

- In this work, the proposed NeAW method is only studied with a specific task (3D learning) and a specific model. To make this learning rule more general, the authors are suggested to also test it on other tasks and model architectures. For example, would it also improve models without the WTA module?



**Summary Of The Paper:**

The authors proposed an unsupervised learning method NeAW Hebbian learning that dynamically switches the learning rule of synapses between Hebbian (H) and anti-Hebbian (aH) depending on the activity of the output neuron. The proposed method is claimed to have reduced neuron activity bias, which leads to the activation of a different subset of neurons for different objects, thus, facilitate better learning. The authors tested the learning algorithm on 3D objects / point cloud recognition tasks and found that compared to Hebbian and anti-Hebbian learning, NeAW training reduces the activity bias of output neurons. On the 3D classification tasks, NeAW Hebbian learning rule also outperforms the existing Hebbian-based rules and end-to-end supervised learning when the data is limited.

**Summary Of The Review:**

This paper proposes a variant of the Hebbian learning rule which uses the output activity to decide if Hebbian or anti-Hebbian learning should be used. The proposed method claimed to have relieved the biased activity problem from existing Hebbian and anti-Hebbian learning rules, thus resulting in better local feature representation for downstream tasks such as 3D object recognition. It needs further validation to be a general unsupervised learning rule.

---

> ### Author Response · Authors · 2022-11-19
> **Response to the Reviewer Sia3 1/3**
>
> We thank the reviewer for their feedback. We hope we have addressed their concerns and questions regarding the paper.
>
> > In the motivation study, the authors identified the biased activity problem with Hebbian-based learning rules and argued that a relaxation of this skewed activity is the key to improve learning. In addition to observing the activities resulting from Hebbian learning, I would also suggest studying if end-to-end learning with better classification performance would result in similar biased activities.
>
> **Reply**: We would like to first clarify that the relation between the balanced activity and the improved performance is assumed in the model with the WTA modules. Note, the WTA modules activate a single neuron for each input point, which makes the activation of output neurons sparse. In other words, the model without the WTA modules does not have the skewed activity unless many output values are suppressed by ReLU function.
>
> Our end-to-end supervised model does not have the WTA modules as they are not differentiable. As the reviewer suggested, we study the neuron activity distribution of the supervised model before and after training. The neuron activity here is the average number of the activation calculated by the same way in Figure 2. We observe that the neuron activity is evenly distributed before training as shown in the new figure (Figure 8(a) in Appendix C). This is an expected result because the random initialization of the weights will randomly activate the neuron through ReLU function. Figure 8(b) displays that the activity is less balanced after training in the supervised model though there is no highly skewed activity observed in Hebbian and anti-Hebbian learning. That is, the balanced activity in the model without the WTA modules does not necessarily indicate the better learning. We add the discussion in Appendix C.

---

> > ### Author Response · Authors · 2022-11-19
> > **Response to the Reviewer Sia3 2/3**
> >
> > > In this work, the proposed NeAW method is only studied with a specific task (3D learning) and a specific model. To make this learning rule more general, the authors are suggested to also test it on other tasks and model architectures. For example, would it also improve models without the WTA module?
> >
> > **Reply**: We would like to note that the learning rule in Equation (1) includes the indicator function. It represents that the weight update is based on the closest input point that is another WTA operation to couple with the WTA modules. Hence, the proposed learning rule is designed along with the WTA modules, which is an important contribution in the paper. Figure 18 in Appendix E is added to show the test accuracy comparison between the model with and without the WTA modules in both ModelNet10 and ModelNet40. The model without the WTA modules achieves 71.8% for ModelNet10 and 25.5% for ModelNet40 while having the WTA modules shows 88.2% for ModelNet10 and 76.2% for ModelNet40. The results show that the model performance significantly drops without the WTA modules. Thus, this ablation study demonstrate that the NeAW learning should accompany the WTA modules. We clarify that the WTA modules play an important role in the section 3 and add the discussion in Appendix E.
> >
> > We also add simple 2D learning (classification) results on point MNIST datasets in Appendix E. The model architecture is same with the other experiments. Each data samples include the point cloud representation of the MNIST images where the coordinates of the dark pixels are included. Figure 19 in Appendix E is added to show that the end-to-end supervised learning achieves 96.3%, and the NeAW learning shows the 94.8%. As the datasets are relatively simple than ModelNet10 and ModelNet40, both the learning methods shows the similarly high accuracy.   The discussion is included in Appendix E.
> >
> > In summary, we demonstrate the results on the point cloud based 2D and 3D datasets. Also, the performance change in the model without the WTA modules is studied. We agree with the reviewer’s view, and further validation with more various model architectures and datasets would be an important future work.

---

> > > ### Author Response · Authors · 2022-11-19
> > > **Response to the Reviewer Sia3 3/3**
> > >
> > > > Some minor typos and unclear phrasing.
> > > Page 2, Equation (1). The  before w(t+1) should be removed.
> > > Page 4, Definition 1. The phrasing is not very clear.
> > >
> > > **Reply**: We apologize for the mistakes in the paper. The Δ in Equation (1) is removed, and Definition 1 is improved.
> > >
> > > >Summary Of The Review:
> > > This paper proposes a variant of the Hebbian learning rule which uses the output activity to decide if Hebbian or anti-Hebbian learning should be used. The proposed method claimed to have relieved the biased activity problem from existing Hebbian and anti-Hebbian learning rules, thus resulting in better local feature representation for downstream tasks such as 3D object recognition. It needs further validation to be a general unsupervised learning rule.
> > >
> > > **Reply**: We appreciate the reviewer’s feedback which helped us to improve the quality of the paper. We also hope that our response and the revised paper have sufficiently addressed the reviewer’s concerns. We particularly agree with the reviewer’s summary; and hence, we are planning to add additional experiments. As the current model architecture is focused on learning point sets, we plan to validate the NeAW learning on other tasks such as the spatiotemporal point cloud prediction and reconstruction. We thank the reviewer again for their time and effort.

---

> > > > ### Comment · Reviewer_Sia3 · 2022-11-21
> > > > **Thanks for making improvements**
> > > >
> > > > I would like to thank the authors for addressing the issues raised and conducting additional experiments to improve the quality of their work. I am satisfied with their improvements and would like to update my recommendation as "8: accept, good paper".

---

> > > > > ### Author Response · Authors · 2022-11-21
> > > > > **Thank you for the positive feedback**
> > > > >
> > > > > We are glad to know that your concerns have been addressed. We would like to kindly remind you to update the score as it is currently 6. Thank you so much again for your positive feedback and effort to improve the quality of our paper.

---

> > > > > > ### Comment · Reviewer_Sia3 · 2022-11-23
> > > > > > **Updated Score**
> > > > > >
> > > > > > Dear authors, I have updated my score for your paper.

---

### Author Response · Authors · 2022-11-19
**Summary of the Revision**

We are grateful to all the reviewers for their time and helpful feedback to improve the quality of our work. Based on their suggestions, we have made the following changes to the main paper:

* Figure 4 is modified to include t-SNE plots for Hebb’s rule, Grossberg’s rule, and Oja’s rule.
* Definition 1 is improved to clearly explain the neuron activity.
* In section 3.2, the main motivation of the paper is additionally explained.
* In section 4 “Balancing of Neuron Activity”, the correlation between the object-dependent activation and improved learning is additionally explained.
* In section 4 “Comparison with other Hebbian learning”, the discussion is improved to emphasize the NeAW Hebbian learning is superior to the others.
* One more baseline (self-supervised learning) is added to Table 2 for comparison, and the memory size of PointNet, PointNet++, and DGCNN are modified (to be classification only models).
* In section 4 “Comparison with Existing Supervised Learning Models”, performance of the supervised and unsupervised learning in the tiny models are discussed.
* The causal relation and correlation between the neuron activity and representation learning are clarified throughout the paper.


The changes made to the appendix include:

* A new Appendix B is included to discuss the causal relation and correlation between the neuron activity and representation learning. Figure 7 is added for this section.
* A new Appendix C is included to discuss the neuron activity distribution of the other learning rules. Figure 8, 9, 10, 11, and 12 are added for this section.
* A new Appendix D is moved from the experimental section. A wider range of hyperparameters in NeAW learning is studied. Figure 14 and 15 are added for this section.
* A new Appendix E is included to discuss the performance in various model architecture including smaller and larger models, self-supervised model, the unsupervised model without WTA. Figure 16, 17, 18, and 19 are added for this section.
* A new Appendix G is included to discuss the biological plausibility of NeAW learning at the perspective of Homeostatic plasticity.

---

### Decision · Program_Chairs · 2023-01-20

**Decision:**

Accept: poster

**Justification For Why Not Higher Score:**

The positives about this paper (mentioned elsewhere) aside, some of the reviewers -- as well as myself -- still feel that the generality of the paper currently is a bit limited. As mentioned, the proposed method has only been tested on point-cloud data (1 dataset in the main paper; and 1 minor dataset added to the Supplementary), even though there is really no good reason that such type of data should have been the first thing to try. Why not even more common data / datasets?

If the algorithm had been shown to work well across several very common datasets, and even across different data types, this would likely have been a spotlight paper, and possibly an oral one.

**Justification For Why Not Lower Score:**

The paper presents a significant contribution at the intersection of biologically-inspired learning and contemporary deep learning, backed by both theoretical and empirical evidence. After some valid initial critiques by reviewers, the authors came back with clear explanations and further empirical evidence to back their claims, and the paper in its current form is much-improved from the initial submission.

**Metareview: Summary, Strengths And Weaknesses:**

This paper proposes a novel unsupervised learning rule called NeAW, which combines both Hebbian and anti-Hebbian learning. It does this by switching between Hebbian (H) and anti-Hebbian (aH) learning, depending on neuronal activity. There are both theoretical proofs and empirical results showing the downside of either type of learning alone, which NeAW can overcome. Specifically, both H and aH learning produce low diversity of neuronal representations, with multiple neurons responding to similar stimuli -- termed activity bias. Acitivity of neurons after NeAW learning are shown to be more diverse. Empirically, NeAW is applied to object classification from 3D point cloud data (with 2D point cloud MNIST data in the Supplementary Material). NeAW is shown to outperform other types of Hebbian-type learning rules, as well as being somewhat comparable to end-to-end supervised learning. In low-data regimes, NeAW outperforms backpropagation as well.


-- STRENGTHS --

1) This is perhaps the first paper (or among the first few) that manages to get Hebbian and anti-Hebbian learning to work well on deep networks and large-scale data

2) Furthermore, the combination of both theoretical proofs and empirical results on relatively large-scale data is another key contribution

3) Strong empirical performance with robust evaluations / comparisons.

4) Overall, a well-written and well-illustrated paper.


-- WEAKNESSES --

1) Empirical results have only been shown on point-cloud data so far, and it's curious why such data was the first choice, since there's no logical link between Hebbian learning and point-cloud data.

2) [Minor] Multiple motivations were given (e.g. biological plausibility, balancing of neural activity, low-data learning, etc.), making it unclear what the key motivation of the paper was.

3) [Minor] The performance compared to fully-supervised learning isn't so great, except in the low-data settings. (That said, it's expected that unsupervised learning methods don't do as well as supervised methods)

**Note From Pc:**

if the above contains the word "oral" or "spotlight" please see: "oral" presentation means -> notable-top-5% and "spotlight" means -> notable-top-25%. As stated in our emails, we are disassociating presentation type from AC recommendations

**Summary Of Ac-Reviewer Meeting:**

All reviewers expressed a significant improvement in their impression of the paper after the authors's responses, with an almost 2-point increase from initial scores. The reviewers felt the authors were very comprehensive in their responses, including sound clarifications, additional analyses such as dissimilarity matrices, as well as clearer evidence showing how the "activity bias" issue of Hebbian / anti-Hebbian learning alone being tackled by their proposed method.

Overall, all reviewers ultimately supported this paper, and confirmed their remaining concerns were fairly minor or "non-fatal" ones. One reviewer in particular (reviewer "rCep") was enthusiastic about the contribution of this paper, being an expert in this area. (Their final rating was an 8, confidence 4).